# Evaluation of the initial timing of infection control pharmacist-driven audit and monitoring of vancomycin therapy in patients with infectious diseases: A retrospective observational study

Hideki Sugita[1,2]*, Natsumi Okada[3], Matoka Okamoto[4], Masakazu Abe[1,3], Masae Sekido[1,5], Michiko Tanaka[1,2], Tatsuro Tamatukuri[1,6], Yuika Naito[1,6], Masayuki Yoshikawa[1,6], Eisuke Inoue[7], Hironori Tanaka[1,8]

1 Department of Hospital Pharmaceutics, Showa University School of Pharmacy, Shinagawa-ku, Tokyo, Japan, 2 Department of Pharmacy, Showa University Fujigaoka Hospital, Yokohama, Kanagawa, Japan, 3 Department of Pharmacy, St. Luke's International Hospital, Chuo-ku, Tokyo, Japan, 4 Division of Clinical Nutrition and Metabolism, Department of Clinical Pharmacy, Showa University School of Pharmacy, Shinagawa-ku, Tokyo, Japan, 5 Department of Pharmacy, Showa University Koto Toyosu Hospital, Koto-ku, Tokyo, Japan, 6 Department of Pharmacy, Showa University Hospital, Shinagawa-ku, Tokyo, Japan, 7 Showa University Research Administration Center, Showa University, Shinagawa-ku, Tokyo, Japan, 8 Department of Pharmacy, Showa University Hospital East Branch, Shinagawa-ku, Tokyo, Japan

* hsugita@cmed.showa-u.ac.jp

**Data Availability Statement:** Concerning data availability, there are restrictions on disclosing individual patients' information for ethical

## Abstract

### Background

Early monitoring and feedback on the treatment of infectious diseases are some of the methods for optimising antimicrobial treatment throughout the treatment period. Prospective audits and feedback interventions have also been shown to improve antimicrobial use and reduce antimicrobial resistance. We examined the appropriate use of antimicrobials by focusing on the initial timing for audits and feedback intervention of antimicrobial prescription by Infection Control Team pharmacists.

### Methods

We conducted a retrospective observational study in a university hospital in Tokyo, Japan from 1 January 2019 to 31 May 2021. We retrospectively enrolled patients with infections and those patients suspected of having an infection, who were administered vancomycin and assessed at our hospital. The definition of primary outcome was the maintenance of target vancomycin trough blood concentrations of 10–20 μg/ml during treatment. Multivariable logistic regression and multivariate linear regression analyses were performed to test the effectiveness of the initial timing of the intervention by Infection Control Team pharmacists as the explanatory variable.

considerations based the recommendation of the Ethics Committee on Research Involving Human Subjects at Showa University. Due to the limited number of patients admitted to our hospital and the small population size at our research centre, only the option of pseudo-anonymization of personal information is available to us. As such, the data set cannot be made available. Contact address: Hideki Sugita (hsugita@cmed.showa-u.ac.jp), Hironori Tanaka (h-tanaka@cmed.showa-u.ac.jp), Matoka Okamoto (matoka.o@cmed.showa-u.ac.jp), Masakazu Abe (k.abe@cmed.showa-u.ac.jp), Masae Skido (m-sekido@cmed.showa-u.ac.jp), Michiko Tanaka (t.michiko@cmed.showa-u.ac.jp), Tatsuro Tamatsukuri (t.tatsuro@cmed.showa-u.ac.jp), Yuika Naito (n-yuika@cmed.showa-u.ac.jp), Masayuki Yoshikawa (masayuki-y@cmed.showa-u.ac.jp), Natsumi Okada (okada.natsumi.r4@luke.ac.jp), Eisuke Inoue (eisuke.inoue@med.showa-u.ac.jp).

**Funding:** Initials of the authors who received each award: HS. Grant numbers awarded to each author: JP20K18942. The full name of each funder: The Japan Society for the Promotion of Science (JSPS) KAKENHI Grant-in-Aid for Early-Career Scientists. URL of each funder website: https://www.jsps.go.jp/english/ Did the sponsors or funders play any role in the study design, data collection and analysis, decision to publish, or preparation of the manuscript? NO. The funders had no role in study design, data collection and analysis, decision to publish, or preparation of the manuscript.

**Competing interests:** I have read the journal's policy and the authors of this manuscript have the following competing interests: EI received lecture fees from Pfizer Japan Inc., Bristol-Myers Squibb Co., and Eisai Co., Ltd., and consulting fees from Nippontect Systems Co., Ltd., and Cyberdine, Inc. The other authors declare no conflicts of interest. This does not alter our adherence to PLOS ONE policies on sharing data and materials.

## Results

A total of 638 patients were included in this study, with a median age of 69 years (interquartile range: 54–78 years). Multivariable logistic regression revealed that the maintenance of target vancomycin trough concentrations was not associated with the timing of the audit and the initiation of monitoring by Infection Control Team pharmacists (adjusted odds ratio: 0.99, 95% confidence interval: 0.99–1.00, p = 0.990). Multivariate linear regression revealed that the duration of vancomycin administration was significantly correlated with the timing of initiation of monitoring by Infection Control Team pharmacists (adjusted estimate: 0.0227, standard error: 0.0051, p = 0.012).

## Conclusions

Our study showed that early initiation of a comprehensive audit and monitoring by Infection Control Team pharmacists did not affect the maintenance of the target vancomycin trough blood concentration. However, it reduced the duration of vancomycin administration.

## Introduction

Infection prevention and control (IPC) is essential for patient safety and quality of care, and the World Health Organization (WHO) has stated key points for providing effective IPC in collaboration with multi-professional healthcare providers [1]. In recent years, the Antimicrobial Stewardship Program (ASP), which promotes the appropriate use of antimicrobials, has focused on the strategic management of antimicrobial use. To combat antimicrobial resistance (AMR), the WHO adopted an Action Plan in 2015 [2], and Japan began working on the AMR Action Plan in 2016 [3]. To promote the appropriate use of antimicrobial agents, the ability to make the right choice of antimicrobial agent is important, and a prerequisite for this is an understanding of the appropriate treatment and control of infectious diseases.

Infectious disease treatment, prevention and control requires a wide range of educational activities, by medical doctors, nurses, pharmacists, and other medical providers. In particular, it is important to provide high-quality education on infectious diseases early on to students through initial training [4, 5]. However, there are limited opportunities to educate healthcare professionals by scheduling separate times for seminars and conferences, and as such, approaches in daily practice might be considered efficient.

Twenty years ago, it was understood that avoiding overuse and inappropriate use of antimicrobials can reduce the development of bacterial resistance [6]. Early monitoring and feedback on the treatment of infectious diseases are some of the methods for optimising antimicrobial treatment throughout the treatment period. These also include proactive monitoring, administering antimicrobial agents, reviewing infectious disease test results, and providing feedback as necessary [7]. Prospective audits and feedback (PAF) interventions also improve antimicrobial use, reduce antimicrobial resistance, and lower *Clostridioides difficile* infection rates without adversely affecting patient outcomes [8–11]. The following are reported as the timing of interventions: at the time it was deemed necessary to alter the choice of an antimicrobial agent or dosage; when the results of infectious disease tests are known; when the effectiveness of treatment is determined; when the route of administration is changed; and when the drug is administered for a long duration [7]. However, direct intervention and feedback by the Infection Control Team (ICT) and Antimicrobial Stewardship Team (AST), whose core members

are infectious disease specialists and clinical pharmacists, are limited in their ability to reach all patients because of time and human resource constraints. The implementation of appropriate antimicrobial use differs between institutions with dedicated ASPs and those without [12]. The timing of AST interventions, which could be daily, thrice a week, or once a week, is tailored to the feasibility of each facility and contributes to the appropriate use of antimicrobials without adversely affecting patient outcomes [8, 10, 11]. Weekly AST interventions reduced antimicrobial use, long-term use rates, drug resistance rates among *Pseudomonas aeruginosa* and MRSA infections, and costs of specific antimicrobials, when compared to those of interventions after a certain period of antimicrobial use (i.e., more than 14 days) [11]. In a clinical pharmacist-led intervention, 70% of the dose and antimicrobial changes were reported to have been accepted following pharmacist-led monitoring and feedback on infectious disease treatment (AST rounds of 1 h each, thrice a week), with no specialist training in infectious diseases [13]. In contrast, a dilemma has been identified with the economic benefits of antimicrobial stewardship (AS) activities, including interactive educational interventions in hospital management, whereby infectious disease specialists spent a considerable amount of time improving clinical effectiveness, i.e., reducing mortality rates and length of hospital stay [8]. A robust ASP required 1.0 full-time pharmacist and 0.25 full-time physician per 100 beds [14]. Therefore, considering the huge labour burden and costs of increasing the number of reviews and PAFs as a practice of ASTs and ICTs, it is important to consider reasonable options with a multi-professional division of labour that utilises expertise from across the board. Among these issues, no reports have examined the appropriate use of antimicrobial drugs with an emphasis on the initial timing to audit and feedback by ICT pharmacists who perform antimicrobial drug logistical audits.

This study assessed the impact of the initial timing of the audit and monitoring intervention of vancomycin by ICT pharmacists, in patients who had developed or were suspected of having an infection, on the maintenance of target vancomycin trough blood concentration during treatment.

## Materials and methods

### Study design and settings

This retrospective, observational study was conducted at an 815-bed university hospital in Eastern Tokyo, Japan. The patients hospitalised from 1 January 2019 to 31 May 2021, were recruited for the study. The follow-up period was from the start of vancomycin treatment to the end of its administration. In this hospital, AST and ICT members collaborated on infectious disease therapy and control issues. The ICT members in this hospital included medical doctors, nurses, clinical microbiologists, and clinical pharmacists who were certified or trained in infection control. The role of ICT pharmacists included reviews of appropriate antimicrobial use, especially anti-methicillin-resistant *Staphylococcus aureus* (MRSA) agents such as vancomycin, teicoplanin, daptomycin, and linezolid, with respect to the suppression of resistant organisms. These reviews were performed once or twice weekly for an hour or two. Further, the ICT pharmacists conducted the audit, provided feedback on the results of reviews and shared them with clinical pharmacists, who provided pharmaceutical care for approximately 50 patients in each hospital ward. If there were several issues, clinical pharmacists negotiated or discussed them with the attending doctors as needed. The ICT pharmacists carried out audits and monitoring of anti-MRSA agents as follows: adequacy of dosage, schedules and treatment duration; recommendation of therapeutic drug monitoring (TDM); implementation of bacterial culture and results; necessity for de-escalation; occurrence of adverse events such

as renal and liver impairment, cytopenia and red man syndrome; and consultation from clinical pharmacists to medical doctors.

## Study population

The eligibility criteria were hospitalised patients who had developed or were suspected of having an infection and were administered intravenous vancomycin. Patients for whom vancomycin was contraindicated or those who did not need to undergo TDM, such as prophylactic administration for surgical site infection, were excluded. Furthermore, if vancomycin administrations were completed before the ICT pharmacists conducted the audit and monitoring of anti-MRSA drug appropriate use, the patients were excluded from the analysis. ICT pharmacists accumulated patient information from 1st March 2019 to 7th April 2021 as a general ICT practice at that time. For this study, investigators retrospectively collected data at baseline from our ICT pharmacists' practical database and electronic medical records from 28th June 2021 to 24th February 2023. The daily ICT practical implementation record in the pharmacy department comprises the ICT practical database. The patient information included in this database was extracted from the electronic medical record. Based on the ICT practical database, the investigators acquired the necessary variable data from each patient's electronic medical records to conduct this investigation.

## Data collection and variables

The primary outcome was the maintenance of target vancomycin trough blood concentration during treatment, whose range was defined from 10 to 20 μg/ml as the trough level. At the time of data collection, a trough value of 10 to 20 μg/ml was recommended as the target concentration of vancomycin, whose range depended on the site and severity of infection, as well as the clinical response [15, 16]. Since this study's population also included complicated infections such as nosocomial pneumonia, meningitis, and bacteraemia [17, 18], the target trough blood concentration range of vancomycin was defined as 10 to 20 μg/ml. If vancomycin trough blood concentration was not measured, the patient was treated as not achieving the target concentration. The secondary outcomes were all-cause mortality within 30 days of starting treatment with vancomycin, duration of vancomycin administration, and implementation of de-escalation without vancomycin.

The initial timing of the audit and monitoring intervention by ICT pharmacists was obtained as continuous variables from our ICT pharmacists' practical database. Pharmacist-led ASP interventions have been implemented once or three times a week in previous studies and have established clinical benefits such as reduced antimicrobial use and shorter treatment times [11, 19]. In our hospital, pharmacist-led patient monitoring of anti-MRSA drug use was conducted once or twice a week during this study period. In this study, the timing of intervention initiation was exploratively categorised into four groups, considering that the intervention timings in previous studies would take place at least once to three times a week as follows: <24 hours, 24 to 72 hours, 72 to 120 hours, ≥120 hours [11, 19]. We retrospectively collected the following data from our ICT pharmacists' practical database and electronic medical records when necessary: 1) the time of the audit and monitoring intervention by ICT pharmacists; diagnosis department; interventions of clinical pharmacists in charge of the wards; vancomycin-related data (duration of administration and blood trough concentration), 2) age; sex; height; weight; comorbidities; immunosuppressant use; intensive care unit (ICU) admission; white blood cells, neutrophils, platelets, creatinine and blood urea nitrogen levels; estimated glomerular filtration rate; aspartate aminotransferase, alanine aminotransferase, γ-glutamyl transpeptidase and albumin levels; concomitant drug use; other vancomycin-related data (the

prescribed date and time, initial dose and dosage); data on diagnoses; suspected infections; the causative or detected organisms. In some cases, general physicians initiated vancomycin administration and performed the concentration measurement order, while in other cases, clinical pharmacists collaborated with general physicians who were involved in the design of the first vancomycin administration strategy, including requests from the ICT pharmacist. Therefore, the timing of the first vancomycin concentration measurement was not consistent for each patient. Estimated creatinine clearance (eCCr) was calculated using the Cockcroft-Gault formula. Three trained ICT pharmacists (HS, MT, and YN) implemented data collection to construct the ICT practical database. In addition, five investigators (HS, NO, MO, KA, and MS) conducted data collection for this study from electronic medical records based on standardised work procedures.

Reasons for vancomycin cessation were classified as cure or improvement, death, hospital discharge, side effects, implementation of de-escalation, or change to other anti-MRSA agents. Adverse events due to vancomycin administration were categorised as renal dysfunction, hearing loss, infusion reaction, skin rash, or allergy.

## Sample size

No previous studies have examined the effect of time on the auditing and monitoring of antimicrobial drugs in pharmaceutical care practices. However, previous before and after observational studies have examined the effect of pharmacist monitoring on achieving optimal blood concentrations in patients using anti-MRSA drugs. The odds ratio was calculated from these results, in which the percentage of patients achieving optimal blood concentrations improved from 31.6% to 59.1% and 27.3% to 55.8% [20, 21]. Furthermore, considering that all cases were implemented in audit and monitoring practices, there was an estimated odds ratio of 2.0 in 2 SD increase in the period (h) between the prescription order for vancomycin and the initial audit and monitoring by the ICT pharmacist, and a correlation coefficient with covariates of 0.6. To detect a difference at a significance level of 5% and power of 90%, 614 cases in both groups were required. Therefore, a target of 650 patients was set, considering dropout.

## Statistical analysis

Regarding the participants' baseline characteristics, we aggregated the median, interquartile range (IQR), counts and percentages. For each variable, an unpaired t-test was used for continuous variables, and Fisher's exact test was used to test categorical variables. Multivariable logistic regression analysis for the primary outcome was performed to test the effectiveness of the periods between the prescription order for vancomycin and the initial audit and monitoring period by the pharmacist in charge of the ICT, adjusted by age, sex, weight, eCCr, creatinine, AST, ALT, albumin, comorbidities (diabetes, cardiovascular diseases, renal failure, hypertension, and dyslipidaemia), immunosuppressant use, ICU admission, vancomycin loading dose, concomitant drug use (aminoglycoside, NSAIDs, piperacillin-tazobactam), and interventions of clinical pharmacists in charge of the wards, at baseline. Confounders that might alter the pharmacokinetics and pharmacodynamics of vancomycin and that might influence the environment in which treatment is received were used to adjust for potential factor differences at baseline. The analyses of secondary outcomes for evaluating the initial timing (h) of the audit and monitoring intervention by the ICT pharmacists were as follows: analysis for death within 30 days was conducted using multivariable logistic regression analysis adjusted by age, sex, eCCr, albumin, comorbidities (cardiovascular diseases), ICU admission, the loading dose of vancomycin, and interventions of clinical pharmacists in charge of the wards at baseline; analysis for implementation of de-escalation was conducted using multivariable logistic regression

analysis adjusted by age, sex, comorbidities (diabetes, cancer, cardiovascular diseases, COPD, hepatic disease, renal failure, hypertension, and dyslipidaemia), immunosuppressant use, ICU admission, and interventions of clinical pharmacists in charge of the wards at baseline; analysis for number of days of vancomycin administration was conducted using multiple linear regression analysis adjusted by age, sex, comorbidities (diabetes, cancer, cardiovascular diseases, COPD, hepatic disease, renal failure, hypertension, and dyslipidaemia), immunosuppressant use, ICU admission, and interventions of clinical pharmacists in charge of the wards at baseline. Subgroup analyses were performed in the strata of paediatric patients, ICU patients, non-ICU patients, and non-haematology patients. Assuming the prophylactic administration of vancomycin for suspected febrile neutropenia, the antimicrobial treatment group in this study was included for evaluation by the department, excluding haematology, as a subgroup analysis. Missing values were imputed using the last-observation-carried-forward method. Furthermore, the missing values for confounding factors required for adjustment were obtained using the mean imputation method. Moreover, explorative analyses of primary and secondary outcomes and the subgroups were performed using the timing of intervention initiation categorised into the four groups (<24 hours, 24 to 72 hours, 72 to 120 hours, ≥120 hours) as explanatory variables. Where there were missing measurements of more than 5%, other imputation methods such as sensitivity analysis were considered. All tests were two-sided, and significance was set at 5%. R version 4.2.1 (The R Foundation for Statistical Computing, Vienna, Austria) was used for statistical analysis.

### Ethical approval statement

This study was approved by the Institutional Review Board of Showa University (number: 406). Only fully anonymised data were used in statistical analyses. As the study was retrospective, deidentified data and patient privacy was protected; thus, the committee waived the need for written informed consent, and each patient could opt out. To protect personal data, the medical information obtained was anonymised by the personal data manager, with a unique symbol and a correspondence table to enable individuals to be identified where necessary. The study was conducted in compliance with the Declaration of Helsinki for protecting the rights and welfare of the patients and followed the Strengthening the Reporting of Observational studies in Epidemiology (STROBE) guidelines for reporting [22].

## Results

### Participants

From the data on ICT practice during the study period (n = 956), 937 patients were eligible and 299 patients were excluded, resulting in 638 patients being included in this study. Patients were excluded due to reasons shown in Fig 1. No patients were allergic to vancomycin.

### Descriptive data

The baseline characteristics are summarised in Table 1. The median age of the patients was 69 years (IQR 54, 78 years). Comorbidities included cancer (56.6%) and cardiovascular disease (46.6%), and renal failure (20.2%). A total of 174 (27.3%) patients were admitted to the ICU. The haematology department frequently used vancomycin for treatment or prophylaxis (36.2%). Piperacillin-tazobactam was administered to 93 patients (14.6%). The numbers of patients for the initial timing of ICT pharmacists' audit and monitoring interventions stratified by the four groups were 138, 279, 136 and 85, respectively. The four groups were comparable

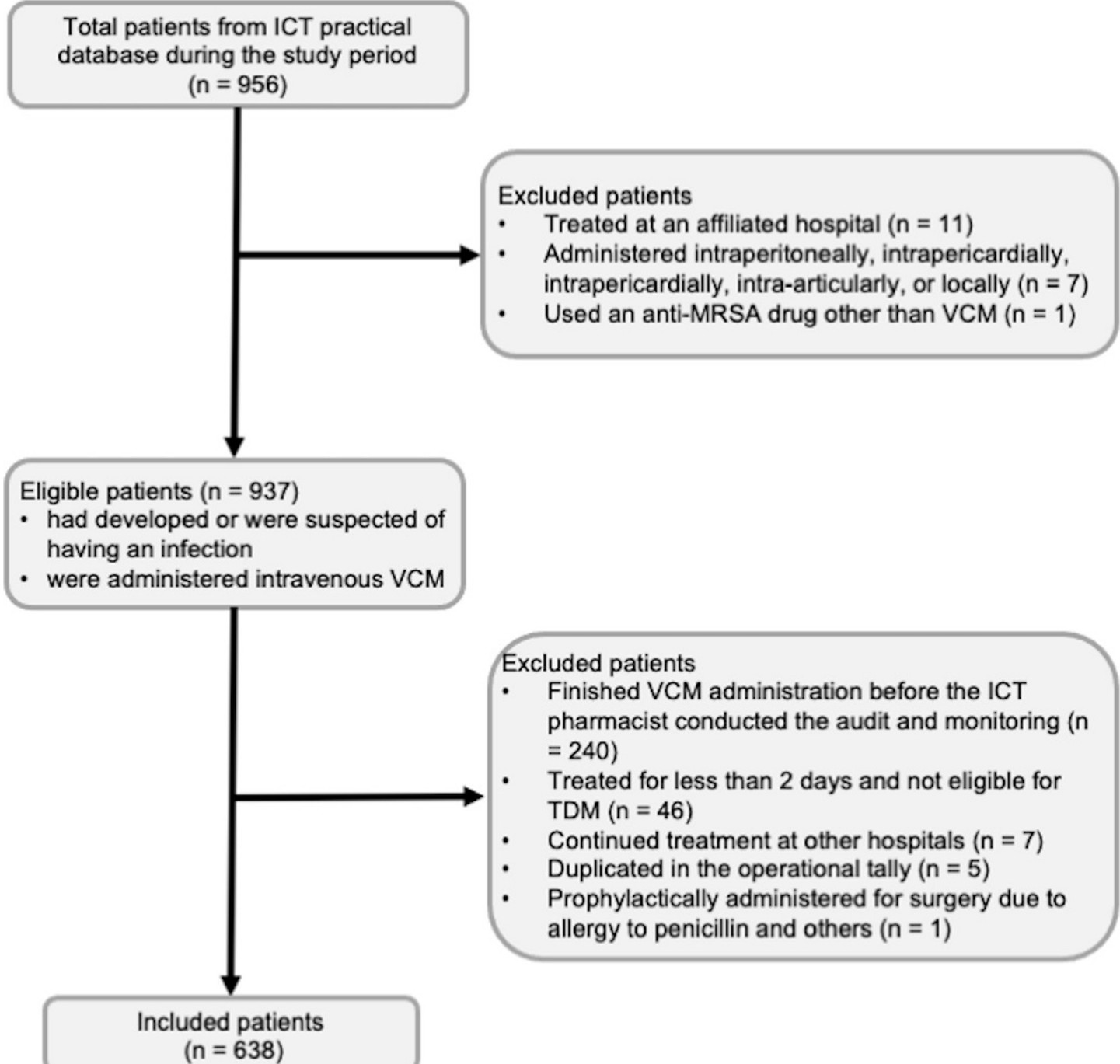

**Fig 1. Patient inclusion.** The baseline period was from 1 January 2019 to 31 March 2021, and the follow-up period was from 1 April 2021 to 30 September 2021. ICT, infection control team; MRSA, methicillin-resistant *Staphylococcus aureus*; TDM, therapeutic drug monitoring; VCM, vancomycin.

in terms of the starting time of ICT pharmacists' audits and monitoring of the variables listed in Table 1.

The median follow-up period for data collection was 30 days (IQR 30, 30). The most frequent diagnoses or suspected infections in patients who were administered vancomycin were

**Table 1. Baseline characteristics of the study population.**

| | Overall (n = 638) | The initial timing of the audit and monitoring intervention by the ICT pharmacists (h) | | | | p-value |
|---|---|---|---|---|---|---|
| | | <24 (n = 138) | ≥24, <72 (n = 279) | ≥72, <120 (n = 136) | ≥120 (n = 85) | |
| Age (year), median [IQR] | 69.0 [54.0, 78.0] | 68.0 [55.2, 77.0] | 69.0 [53.0, 78.0] | 69.0 [55.8, 79.2] | 71.0 [61.0, 77.0] | 0.730 |
| Sex, male, n (%) | 229 (35.9) | 48 (34.8) | 104 (37.3) | 49 (36.0) | 28 (32.9) | 0.890 |
| Height (cm), median [IQR] | 162.0 [153.8, 167.2] | 161.5 [153.0, 167.9] | 162.0 [154.0, 167.0] | 161.2 [152.4, 167.0] | 163.0 [158.0, 168.0] | 0.467 |
| Weight (kg), median [IQR] | 52.7 [44.8, 63.1] | 52.3 [44.4, 62.7] | 53.5 [46.5, 63.4] | 52.6 [43.7, 62.4] | 51.7 [45.3, 64.3] | 0.857 |
| Comorbidity, n (%) | | | | | | |
| Diabetes | 200 (31.3) | 46 (33.3) | 85 (30.5) | 46 (33.8) | 23 (27.1) | 0.690 |
| Cancer | 361 (56.6) | 84 (60.9) | 151 (54.1) | 71 (52.2) | 55 (64.7) | 0.167 |
| Cardiovascular diseases | 297 (46.6) | 62 (44.9) | 126 (45.2) | 73 (53.7) | 36 (42.4) | 0.291 |
| COPD | 50 (7.8) | 8 (5.8) | 17 (6.1) | 14 (10.3) | 11 (12.9) | 0.104 |
| Hepatic disease | 129 (20.2) | 22 (15.9) | 58 (20.8) | 36 (26.5) | 13 (15.3) | 0.103 |
| Renal failure | 129 (20.2) | 31 (22.5) | 52 (18.6) | 27 (19.9) | 19 (22.4) | 0.774 |
| Hypertension | 250 (39.2) | 53 (38.4) | 108 (38.7) | 52 (38.2) | 37 (43.5) | 0.853 |
| Dyslipidaemia | 121 (19.0) | 27 (19.6) | 56 (20.1) | 24 (17.6) | 14 (16.5) | 0.861 |
| Immunosuppressant use, n (%) | 173 (27.1) | 42 (30.4) | 70 (25.1) | 38 (27.9) | 23 (27.1) | 0.707 |
| ICU duration of stay, n (%) | 174 (27.3) | 35 (25.4) | 80 (28.7) | 41 (30.1) | 18 (21.2) | 0.442 |
| Diagnosis department, n (%) | | | | | | |
| Non-haematology | 407 (63.8) | 84 (60.9) | 175 (62.7) | 91 (66.9) | 57 (67.1) | 0.663 |
| Haematology | 231 (36.2) | 54 (39.1) | 104 (37.3) | 45 (33.1) | 28 (32.9) | |
| Intervention of clinical pharmacist in the ward, n (%) | 275 (43.1) | 55 (39.9) | 118 (42.3) | 60 (44.1) | 42 (49.4) | 0.554 |
| WBC (10^3/μl), median [IQR] | 8.2 [2.7, 13.0] | 8.2 [2.1, 12.4] | 8.6 [2.2, 13.6] | 8.3 [3.1, 12.4] | 7.0 [3.5, 12.4] | 0.923 |
| Neutrophil (/μl), median [IQR] | 6990.0 [2325.0, 11215.0] | 6725.0 [2235.0, 11020.0] | 7390.0 [2410.0, 11625.0] | 6970.0 [2305.0, 10790.0] | 6220.0 [2652.5, 11167.5] | 0.899 |
| Platelet (10^4/μl), median [IQR] | 13.5 [3.7, 24.7] | 12.5 [3.7, 23.1] | 13.3 [3.8, 24.5] | 14.4 [4.0, 27.9] | 13.7 [4.4, 23.2] | 0.556 |
| Creatinine (mg/dl), median [IQR] | 0.7 [0.5, 1.1] | 0.8 [0.5, 1.3] | 0.7 [0.5, 1.1] | 0.7 [0.5, 1.2] | 0.7 [0.6, 1.0] | 0.626 |
| BUN (mg/dl), median [IQR] | 21.0 [13.1, 34.2] | 24.9 [14.4, 37.9] | 20.3 [12.7, 33.1] | 19.4 [12.3, 32.2] | 20.7 [13.7, 35.3] | 0.083 |
| eGFR (ml/min/1.73m$^2$), median [IQR] | 72.0 [42.0, 105.5] | 68.2 [37.4, 106.8] | 77.2 [46.5, 107.5] | 70.8 [39.7, 95.3] | 74.5 [47.9, 104.7] | 0.598 |
| AST (u/l), median [IQR] | 25.0 [16.0, 43.0] | 25.0 [16.0, 42.5] | 25.5 [17.0, 43.8] | 28.0 [16.0, 45.0] | 23.0 [16.0, 40.0] | 0.645 |
| ALT (u/l), median [IQR] | 24.0 [13.0, 42.0] | 22.5 [13.0, 42.5] | 23.0 [13.5, 41.5] | 25.0 [13.0, 44.2] | 25.0 [14.0, 38.0] | 0.917 |
| γ-GTP (u/l), median [IQR] | 60.0 [32.0, 129.8] | 58.5 [30.0, 122.0] | 65.0 [31.0, 141.0] | 54.0 [34.0, 124.0] | 57.0 [39.0, 117.5] | 0.813 |
| Total bilirubin (mg/dl), median [IQR] | 0.7 [0.5, 1.1] | 0.6 [0.4, 0.9] | 0.7 [0.5, 1.2] | 0.7 [0.5, 1.0] | 0.6 [0.5, 0.9] | 0.096 |
| Albumin (g/dl), median [IQR] | 2.7 [2.2, 3.1] | 2.7 [2.2, 3.2] | 2.7 [2.3, 3.1] | 2.7 [2.3, 3.2] | 2.7 [2.2, 3.1] | 0.871 |
| eCCr (ml/min), median [IQR] | 63.7 [36.7, 103.0] | 63.7 [35.3, 100.5] | 72.0 [39.2, 103.5] | 59.2 [35.5, 101.6] | 60.6 [37.3, 103.3] | 0.611 |
| Loading dose of vancomycin, n (%) | 202 (31.7) | 38 (27.5) | 94 (33.7) | 40 (29.4) | 30 (35.3) | 0.484 |
| Concomitant drug, n (%) | | | | | | |
| Aminoglycoside | 21 (3.3) | 4 (2.9) | 9 (3.2) | 4 (2.9) | 4 (4.7) | 0.883 |
| Colistin | 0 (0) | 0 (0) | 0 (0) | 0 (0) | 0 (0) | NA |
| NSAIDs | 46 (7.2) | 9 (6.5) | 14 (5.0) | 13 (9.6) | 10 (11.8) | 0.119 |
| Piperacillin | 1 (0.2) | 0 (0.0) | 1 (0.4) | 0 (0.0) | 0 (0.0) | 0.732 |

(*Continued*)

**Table 1.** (Continued)

| | Overall (n = 638) | The initial timing of the audit and monitoring intervention by the ICT pharmacists (h) | | | | p-value |
|---|---|---|---|---|---|---|
| | | <24 (n = 138) | ≥24, <72 (n = 279) | ≥72, <120 (n = 136) | ≥120 (n = 85) | |
| Piperacillin-tazobactam | 93 (14.6) | 13 (9.4) | 48 (17.2) | 23 (16.9) | 9 (10.6) | 0.103 |

Missing data at baseline were neutrophil count (n = 59), AST (n = 1), γ-GTP (n = 40), bilirubin (n = 9), albumin (n = 2), eGFR (n = 36) and eCCr (n = 31). Missing values of eGFR and eCCr were included in patients <18 years (n = 34).

ICT, infection control team; IQR, interquartile range; COPD, chronic obstructive pulmonary disease; ICU, intensive care unit; WBC, white blood cell; BUN, blood urea nitrogen; eGFR, estimated glomerular filtration rate; AST, aspartate aminotransferase; ALT, alanine aminotransferase; γ-GTP, γ-glutamyl transpeptidase; eCCr, estimate creatinine clearance; NSAIDs, non-steroidal anti-inflammatory agents.

bacteraemia (25.2%), pneumonia (18.5%) and febrile neutropenia (14.1%) (S1 Table). The organisms commonly detected by microbiological culture in vancomycin-treated patients were methicillin-resistant coagulase-negative *Staphylococcus* spp. (16.5%) and MRSA (13.9%) (S2 Table). Either cure or improvement was observed in 37.5% of the patients, and implementation of de-escalation was observed in 32.3% as the reason for the cessation of vancomycin (Table 2). Common adverse events caused by vancomycin occurred in 72 patients and included those with renal impairment (44, 6.9%) and skin rash or allergy (28, 4.7%). Of these, vancomycin treatment was discontinued in 46 (7.2%) patients, but in those with deafness and red man syndrome, vancomycin treatment was not discontinued (Table 2).

## Outcome data

**Primary outcome.** Maintenance of target vancomycin trough blood concentration during treatment was achieved in 307 (48.1%) patients in each patients' treatment period (Table 3). The four exploratory classification groups were as follows: <24 h (67, 10.5%), ≥24 h, <72 h (137, 21.5%), ≥72 h, <120 h (64, 10%), and ≥120 (39, 6.1%) (S3 Table). The adjusted odds ratio (OR) from the multivariable logistic regression for the maintenance of target vancomycin trough blood concentration during treatment was 0.99 (unadjusted OR 1.00, 95% confidence interval (CI) 0.99–1.00, p = 0.990) (Table 3 and S1 Fig). There was no significant difference

**Table 2. Reasons for the cessation of vancomycin treatment and adverse events of vancomycin.**

| | Overall (N = 638) | The initial timing of the audit and monitoring intervention by the ICT pharmacists (h) | | | |
|---|---|---|---|---|---|
| | | <24 (n = 138) | ≥24, <72 (n = 279) | ≥72, <120 (n = 136) | ≥120 (n = 85) |
| Reason for cessation of vancomycin, n (%) | | | | | |
| Cure or an improvement | 239 (37.5) | 37 (26.8) | 103 (36.9) | 59 (43.4) | 40 (47.1) |
| Death | 34 (5.3) | 9 (6.5) | 18 (6.5) | 5 (3.7) | 2 (2.4) |
| Hospital discharge [a] | 39 (6.1) | 7 (5.1) | 16 (5.7) | 11 (8.1) | 5 (5.9) |
| Side effects | 46 (7.2) | 12 (8.7) | 17 (6.1) | 11 (8.1) | 6 (7.1) |
| Implementation of de-escalation | 206 (32.3) | 49 (35.5) | 92 (33.0) | 39 (28.7) | 26 (30.6) |
| Change to other anti-MRSA agents | 74 (11.6) | 24 (17.4) | 33 (11.8) | 11 (8.1) | 6 (7.1) |
| Adverse events caused by vancomycin, n (%) | | | | | |
| Renal impairment | 44 (6.9) | 12 (8.7) | 17 (6.1) | 6 (4.4) | 9 (10.6) |
| Skin rash or allergy | 28 (4.7) | 11 (7.9) | 7 (2.5) | 9 (6.6) | 3 (3.6) |
| None | 564 (88.4) | 115 (83.3) | 255 (91.4) | 121 (89.0) | 73 (85.9) |

[a]Hospital discharge included transfers to hospitals. ICT, infection control team; MRSA, methicillin-resistant *Staphylococcus aureus*.

**Table 3. Primary and secondary outcomes for vancomycin administration based on the initial timing (h) of the audit and monitoring intervention by the ICT pharmacists.**

| Outcomes | No. of patients with events (%) | Multivariable logistic regression | | |
|---|---|---|---|---|
| | | Adjusted OR | 95% CI | p-value |
| Primary outcome (n = 638) | | | | |
| Maintenance of target VCM trough blood concentration during treatment [a], yes | 307 (48.1) | 0.99 | 0.99–1.00 | 0.990 |
| Secondary outcomes (n = 638) | | | | |
| Death within 30 days [b], yes | 90 (14.1) | 0.99 | 0.99–1.00 | 0.106 |
| Implementation of de-escalation [c], yes | 206 (32.3) | 0.99 | 0.99–1.00 | 0.279 |
| | | Multiple linear regression | | |
| | Average [SD] | Estimates | SE | p-value |
| Number of days of VCM administration [a], day | 11.3 [7.4] | 0.0227 | 0.0051 | 0.012 |

The main explanatory variables were the following continuous variables; the initial timing of the audit and monitoring intervention by the ICT pharmacists (h).

[a]The odds ratios derived from the multivariable logistic regression or the estimates from the multiple linear regression were adjusted for age, sex, weight, eCCr, creatinine, AST, ALT, albumin, comorbidities (diabetes, cardiovascular diseases, renal failure, hypertension and dyslipidaemia), immunosuppressant use, ICU admission, vancomycin loading dose, concomitant drug use (aminoglycoside, NSAIDs and piperacillin-tazobactam), and interventions of clinical pharmacists in charge of the wards.

[b]The odds ratios derived from the multivariable logistic regression were adjusted for age, sex, eCCr, albumin, comorbidities (cardiovascular diseases), ICU admission, the loading dose of vancomycin, and interventions of clinical pharmacists in charge of the wards.

[c]The odds ratios derived from the multivariable logistic regression were adjusted for age, sex, comorbidities (diabetes, cancer, cardiovascular diseases, COPD, hepatic disease, renal failure, hypertension and dyslipidaemia), immunosuppressant use, ICU admission, and interventions of clinical pharmacists in charge of the wards.

ICT, infection control team; OR, odds ratio; 95% CI, 95% confidence interval; SD, standard deviation; SE, standard error; VCM, vancomycin; eCCr, estimate creatinine clearance; AST, aspartate aminotransferase, ALT, alanine aminotransferase; ICU, intensive care unit; NSAIDs, non-steroidal anti-inflammatory agents; COPD, chronic obstructive pulmonary disease.

between the intervention's initial timing (<24 h) categorised by the four exploratory classification groups as a reference and other categories (S3 Table). For the adjusted covariates, intrinsic factors affecting the pharmacokinetics of vancomycin, such as renal function and comorbidities, and extrinsic factors, such as the patient's environment, including the ICU setting, were selected. Vancomycin was administered for a median of 10 days (IQR 6.0, 14.0).

**Secondary outcomes.** The adjusted OR from the multivariable logistic regression for death within 30 days after administering vancomycin and implementation of de-escalation of vancomycin to narrow-spectrum were 0.99 and 0.99 (unadjusted OR 0.99, 95% CI 0.99–1.00, p = 0.106; unadjusted OR 0.99, 95% CI 0.99–1.00, p = 0.279), respectively (Table 3 and S1 Fig). The intervention's initial timing <24 h categorised by the four groups as a reference and other categories was comparable (S3 Table). The adjusted estimate from the multiple linear regression for the days of vancomycin administration was related to the intervention start time as a continuous variable (unadjusted estimates, 0.0231; adjusted estimates, 0.0227; SE, 0.0051; p = 0.012) (Table 3 and S1 Fig). Furthermore, compared with the intervention starting time of <24 h as a reference, the intervention's initial timing of ≥120 h only increased the days of vancomycin administration (unadjusted estimates, 3.192, adjusted estimates 3.120; 1.026, p = 0.165) (S3 Table).

## Subgroup analysis

As comparable results, significant relations were derived for the days of vancomycin administration from multiple regression analysis in the population aged ≥18, the non-ICU setting and the department of non-haematology (unadjusted estimates, 0.0232, adjusted estimates, 0.0231, SE, 0.005, p = 0.019; unadjusted estimates, 0.0273, adjusted estimates, 0.0272, SE, 0.006,

p = 0.026; unadjusted estimates, 0.0173, adjusted estimates, 0.0154, SE, 0.007, p = 0.030; Table 4). The results of other multivariable logistic regression analyses were also comparable in the subgroup populations (S4 Table). Moreover, the adjusted OR from the multivariable logistic regression for the implementation of the de-escalation of vancomycin was 0.41 (unadjusted OR, 0.43; 95% CI 0.19–0.91, p = 0.028) in which the intervention starting time of ≥120 h compared with the intervention's initial timing of <24 h as a reference in the non-haematology department (S5 Table). S5 Table shows the results of other subgroup analyses for vancomycin administration based on the initial timing (h) of the audit and monitoring intervention by the ICT pharmacists categorised into the four groups.

## Sensitivity analysis

Sensitivity analyses were not performed because there were few missing values for the confounders required for adjustment (Table 1).

## Discussion

We evaluated the impact of a retrospective audit and feedback targeting the anti-MRSA agent, vancomycin, in a single urban setting at a university hospital. Despite differences in patient populations, such as different departments commonly making the diagnoses and critical care settings, there was no significant correlation between the timing of the audit and the initiation of monitoring by ICT pharmacists and the maintenance of target vancomycin trough blood concentrations. There was a significant association between the number of days of vancomycin administration and the timing of the audit and monitoring initiation by the ICT pharmacists. Comparable results were derived from different populations, such as those aged ≥18, non-ICU settings and non-haematology departments. Furthermore, the number of days of vancomycin administration tended to increase after 120 h compared to the timing of the intervention, up to <24 h. There were no significant differences based on the timing of intervention

**Table 4. Subgroup analysis for number of days of VCM administration based on the initial timing (h) of the audit and monitoring intervention by the ICT pharmacists.**

| Outcomes | Average [SD] | Multiple linear regression | | |
| --- | --- | --- | --- | --- |
| | | Estimates | SE | p-value |
| Age ≥18 years (n = 604) [a], day | 11.4 [7.5] | 0.023 | 0.005 | 0.019 |
| ICU setting, yes (n = 174) [b], day | 10.5 [7.1] | 0.007 | 0.011 | 0.666 |
| Non-ICU setting, yes (total n = 464) [a], day | 11.6 [7.5] | 0.027 | 0.006 | 0.026 |
| Non-haematology, yes (n = 407) [a], day | 11.1 [7.5] | 0.015 | 0.007 | 0.030 |

The main explanatory variables were the following continuous variables; the initial timing of the audit and monitoring intervention by the ICT pharmacists (h).

[a]The estimate from the multiple linear regression was adjusted for age, sex, weight, eCCr, creatinine, AST, ALT, albumin, comorbidities (diabetes, cardiovascular diseases, renal failure, hypertension and dyslipidaemia), immunosuppressant use, ICU admission, vancomycin loading dose, concomitant drug use (aminoglycoside, NSAIDs and piperacillin-tazobactam), and interventions of clinical pharmacists in charge of the wards.

[b]The estimate from the multiple linear regression was adjusted for age, sex, weight, eCCr, creatinine, AST, ALT, albumin, comorbidities (diabetes and cardiovascular diseases), ICU admission, loading dose of vancomycin, concomitant drug use (aminoglycoside, NSAIDs and piperacillin-tazobactam), and interventions of clinical pharmacists in charge of the wards.

Numbers [median (IQR)] of the days of VCM administration for the populations aged ≥18 years, ICU setting, non-ICU setting and non-haematology were 10.0 (6.0, 14.0), 9.0 (6.0, 14.0), 10.0 (6.0, 14.25) and 10.0 (6.0, 14.0), respectively.

ICT, infection control team; OR, odds ratio; 95% CI, 95% confidence interval; SD, standard deviation; SE, standard error; VCM, vancomycin; eCCr, estimate creatinine clearance; AST, aspartate aminotransferase, ALT, alanine aminotransferase; ICU, intensive care unit; NSAIDs, non-steroidal anti-inflammatory agents; COPD, chronic obstructive pulmonary disease; IQR, interquartile range.

initiation in deaths within 30 days of vancomycin administration or vancomycin de-escalation.

There was no association between the period from initial vancomycin prescription to the audit and monitoring by ICT pharmacists, and whether the target vancomycin trough blood concentration was maintained. The effectiveness of vancomycin monitoring by clinical pharmacists has been reported to improve the achievement rate of an appropriate vancomycin trough blood concentration [20, 21]. In these reports, clinical pharmacists associated with the wards checked individual patient parameters after vancomycin administration and monitored the patients for renal function, other laboratory values, and vancomycin levels daily on weekdays or at least twice a week, if necessary, while vancomycin therapy was continued [20, 21]. It may be difficult to ascertain and modify the trends in the narrow therapeutic windows of blood concentrations of vancomycin, which should be strictly controlled for all parameters of individual patients [23, 24], based on the limited time spent on audit and monitoring practices by ICT pharmacists.

During the study period, only 43.1% of the clinical pharmacists were involved in designing the first dose of vancomycin and suggesting the timing of trough concentration measurements, before prescription of vancomycin by the physicians (Table 1). There may have been a lack of adequate dosing and monitoring from the start of vancomycin treatment, although multivariate analysis was performed by adjusting the implementation of initial intervention by clinical pharmacists. There may be a need to strengthen and consider an additional system of collaboration between clinical pharmacists and ICT pharmacists. Overall, 31.7% of the patients were treated with an initial vancomycin loading dose. In the guidelines proposed during the study period, the loading dose might be considered, in order to rapidly achieve the target concentrations in seriously ill patients [15, 16]. We may not have been able to appropriately evaluate whether the patient was critically ill or not, since we had not assessed the severity of the disease in each patient in our general practice in this investigation. Only 48.1% of all the patients maintained their target vancomycin trough blood concentration during therapy. The loading dose may not have been administered to all critically ill patients as recommended by the guidelines, preventing them from reaching the initial target concentration. A systematic review without a meta-analysis indicated that in patients with haematological malignancies or neutropenia, conventional vancomycin dosing results in suboptimal concentrations [25]. Therefore, the intervention start time did not affect the achievement of target blood concentrations, although haematologic patients, who also might be included in prophylactic administration, were excluded from the subgroup analysis. However, it might also be necessary to ensure the safety of clinical practice in the early stages because a decrease in the incidence of nephrotoxicity has been reported in immunocompromised febrile patients with haematologic malignancies, by monitoring serum vancomycin concentration [26].

In this study, an association was found between an increase in the time to initiation of appropriate monitoring of antimicrobial use by ICT pharmacists and an increase in the number of days of vancomycin administration. The effectiveness of multidisciplinary professional interventions in the treatment of infections with antimicrobial agents has been reported as a reduction in antimicrobial days of therapy (6% or 64%, as reported in two studies), total antimicrobial expenditure (37%) and antimicrobial use (event n/total n = 10/104) [19, 27]. Moreover, a decrease in nosocomial infections caused by *Clostridioides difficile* and the emergence of resistant bacteria, such as *Enterobacteriaceae* and vancomycin-resistant *Enterococci* (7.5%), were related to a decrease in the use of antimicrobials [8, 9, 28]. The results of systematic reviews, whose meta-analysis was performed to include randomised trials and uncontrolled observational studies, including ICU patients, indicated that there was no difference in mortality, although antimicrobial use or prescribing patterns were reduced before and after the

implementation of audits and feedback as AST [29, 30]. Adverse drug events (ADEs) occurred with the use of broad-spectrum antimicrobials, including vancomycin [31, 32], and the risk of ADEs increased by 3% very 10 additional days of antimicrobial therapy [32]. The results of previous studies and this study on the association between those interventions and the number of days of antimicrobial use were consistent. This may suggest the need for early intervention.

After the implementation of ASP, broad-spectrum antimicrobial use was reduced to 10% and 17% in ICU and non-ICU settings, respectively, particularly for cephalosporin and glyco-peptide use [33]. From our results in non-haematological populations, the audit and monitoring interventions had a positive impact on the de-escalation implementation, similar to the findings of a previous study [33]. De-escalation to earlier interventions was reduced after > 5 days in the non-haematology population. The early timing of the audit and feedback interventions may have influenced physicians' prescription practices. There was no increase in deaths within 30 days of initiating prescriptions. Even if the intervention by audit and monitoring followed by de-escalation had begun earlier, safety could have been secured.

In this study, 51.9% of the patients failed to maintain vancomycin trough blood concentrations in the range of 10 to 20 μg/ml, and 6.9% of all patients had renal dysfunction or were suspected to be affected by or renal dysfunction due to vancomycin. The frequency of acute kidney injury by vancomycin in previous studies in patients with baseline serum creatinine levels below 2.0 mg/dl was 5%; for concentrations <10 μg/ml, 21%; for 10 to 15 μg/ml, 20%; for 15 to 20 μg/ml; and for >20 μg/ml, 33% [34]. Although 20.2% of the patients in this study had abnormal renal function at baseline, the frequency of developing renal dysfunction was not higher than that reported in previous studies, and the intervention methods of ICT pharmacists in this study may not have worsened renal function. Although we adjusted for variables reflecting renal and hepatic function when conducting the multivariate analysis, it was possible that these variables may have fluctuated after the initiation of vancomycin treatment. Further analysis over time may be necessary to verify the association between strict vancomycin concentration trends and laboratory values affecting these concentrations. A possible side effect of vancomycin was skin rash or allergy (4.7%). Although common side effects include phlebitis and ototoxicity, they were not reported in this study and did not affect treatment performance. However, ototoxicity is often difficult to detect and may require careful monitoring in older adult patients who are at high risk.

This study had some limitations. First, as this was a single-centre study conducted in an urban university hospital with a medically complex patient population, the generalisations based on this study are limited. Second, issues related to potential confounders and selection bias may have affected the validity of the findings. However, the sufficient sample size obtained in our study allowed the analysis of factors affecting the blood concentration of vancomycin to be adjusted as much as possible. Furthermore, the selection bias at baseline was likely small because the ICT practical database utilised in this study was linked to the prescription ordering system and included all patients prescribed vancomycin. Third, there is a possibility of overestimating the effectiveness of the timing of audit and monitoring interventions. This might reflect not only the effect of the ICT pharmacist intervention but also the effect of interventions by other professionals (including physicians, nurses, and laboratory microbiologists), such as AST. However, based on the combination of the ICT pharmacists' practice and AST, benefits and synergies could be expected. Lastly, owing to changes in pharmacists' work needs from that of other healthcare providers and staffing issues during the study period in our hospital, the frequency of intervention may not have remained constant during the vancomycin treatment period, even though audits and monitoring were conducted at least once or twice weekly. To verify the effectiveness of the intervention, future analyses should include the number of interventions per week as in previous studies [10, 11].

The finding of this study showed that the number of days of administration tended to increase with any delay in the timing of early ICT pharmacist intervention. The intervention methods of ICT pharmacists may ensure the safety of vancomycin administration, as there were reports that showed a progressive increase in nephrotoxicity with an increase in the duration of vancomycin treatment [35]. Moreover, the safety of the intervention may be ensured, as no increase in mortality was observed. However, as audits and monitoring are time-consuming and labour-intensive, it is important to maintain IPCs to combat AMR in healthcare facilities while ensuring the availability of healthcare resources. The 2020 guideline has recommended area under the concentration time curve (AUC)-based dosing for the efficacy and safety of vancomycin, and the 2022 Japanese guidelines have also been revised [36, 37]. To adhere to strict blood collection times for the accurate assessment of vancomycin concentrations, and implement the treatment as early as possible considering the healthcare resources, future perspective strategies may need to be explored, so as to further strengthen collaboration for shared specialised knowledge between ICT pharmacists and general ward pharmacists.

## Conclusion

The initial timing of the comprehensive audit and monitoring intervention by ICT pharmacists did not affect the maintenance of the target blood concentration of vancomycin. However, an earlier start of the intervention was associated with a time-dependent reduction in the number of days of vancomycin administration.

## Supporting information

**S1 Table. Proportion of diagnosed or suspected infections in patients administered vancomycin (n = 638).**
(XLSX)

**S2 Table. Organisms implicated or detected in patients treated with vancomycin (n = 638).**
MRCNS, Methicillin-resistant coagulase-negative *Staphylococcus* spp.
(XLSX)

**S3 Table. Primary and secondary outcomes for vancomycin administration based on the initial timing (h) of the audit and monitoring intervention by the ICT pharmacists, categorised exploratively into four groups.** The main explanatory variables were the initial timing of the audit and monitoring intervention by the ICT pharmacists (h) categorised as follows: <24 h, 24 to 72 h, 72 to 120 h, ≥120 h. [a]The odds ratios derived from the multivariable logistic regression or the estimates from the multiple linear regression were adjusted for age, sex, weight, eCCr, creatinine, AST, ALT, albumin, comorbidities (diabetes, cardiovascular diseases, renal failure, hypertension, and dyslipidaemia), immunosuppressant use, ICU admission, vancomycin loading dose, concomitant drug use (aminoglycoside, NSAIDs, piperacillin-tazobactam), and interventions of clinical pharmacists in charge of the wards. [b]The odds ratios derived from the multivariable logistic regression were adjusted for age, sex, eCCr, albumin, comorbidities (cardiovascular diseases), ICU admission, the loading dose of vancomycin, and interventions of clinical pharmacists in charge of the wards. [c]The odds ratios derived from the multivariable logistic regression were adjusted for age, sex, comorbidities (diabetes, cancer, cardiovascular diseases, COPD, hepatic disease, renal failure, hypertension, and dyslipidaemia), immunosuppressant use, ICU admission, and interventions of clinical pharmacists in charge of the wards. ICT, infection control team; OR, odds ratio; 95% CI, 95% confidence interval; SD, standard deviation; SE, standard error; VCM, vancomycin; eCCr, estimate creatinine clearance; AST, aspartate aminotransferase, ALT, alanine aminotransferase; ICU,

intensive care unit; NSAIDs, non-steroidal anti-inflammatory agents; COPD, chronic obstructive pulmonary disease.
(XLSX)

**S4 Table. Results of subgroup analyses for vancomycin administration based on the initial timing (h) of the audit and monitoring intervention by the ICT pharmacists, excluding data presented in Table 4.** The main explanatory variables were the following continuous variables: the initial timing of the audit and monitoring intervention by the ICT pharmacists (h). [a]The odds ratios derived from the multivariable logistic regression were adjusted for age, sex, weight, eCCr, creatinine, AST, ALT, albumin, comorbidities (diabetes, cardiovascular diseases, renal failure, hypertension and dyslipidaemia), immunosuppressant use, ICU admission, vancomycin loading dose, concomitant drug use (aminoglycoside, NSAIDs and piperacillin-tazobactam) and interventions of clinical pharmacists in charge of the wards. [b]The odds ratios derived from the multivariable logistic regression were adjusted for age, sex, eCCr, albumin, ICU admission, vancomycin loading dose and interventions of clinical pharmacists in charge of the wards. [c]The odds ratios derived from the multivariable logistic regression were adjusted for age, sex, comorbidities (diabetes, cancers, cardiovascular diseases, COPD, hepatic diseases, renal failure, hypertension, and dyslipidaemia), immunosuppressant use, ICU admission and interventions of clinical pharmacists in charge of the wards. [d]The odds ratios derived from the multivariable logistic regression were adjusted for age, sex, eCCr, albumin, vancomycin loading dose, concomitant drug use (piperacillin-tazobactam) and interventions of clinical pharmacists in charge of the wards. [e]The odds ratios derived from the multivariable logistic regression were adjusted for age, sex and eCCr. [f]The odds ratios derived from the multivariable logistic regression were adjusted for age, sex, comorbidities (diabetes, cancers, and cardiovascular diseases) and interventions of clinical pharmacists in charge of the wards. [g]The odds ratios derived from the multivariable logistic regression were adjusted for age, sex, eCCr and interventions of clinical pharmacists in charge of the wards. [h]The odds ratios derived from the multivariable logistic regression were adjusted for age, sex, comorbidities (diabetes, cancers, cardiovascular diseases, COPD, hepatic diseases, renal failure, hypertension, and dyslipidaemia), immunosuppressant use and interventions of clinical pharmacists in charge of the wards. [i]The odds ratios derived from the multivariable logistic regression were adjusted for age, sex, eCCr, ICU admission and interventions of clinical pharmacists in charge of the wards. [j]The odds ratios derived from the multivariable logistic regression were adjusted for age, sex, comorbidities (diabetes, cancer, cardiovascular diseases, COPD, hepatic disease, renal failure, hypertension, and dyslipidaemia), immunosuppressant use, ICU admission, and interventions of clinical pharmacists in charge of the wards. ICT, infection control team; OR, odds ratio; 95% CI, 95% confidence interval; SE, standard error; VCM, vancomycin; eCCr, estimate creatinine clearance; AST, aspartate aminotransferase, ALT, alanine aminotransferase; ICU, intensive care unit; NSAIDs, non-steroidal anti-inflammatory agents; COPD, chronic obstructive pulmonary disease; IQR, interquartile range.
(XLSX)

**S5 Table. Results of subgroup analyses for vancomycin administration based on the initial timing (h) of the audit and monitoring intervention by the ICT pharmacists, categorised exploratively into four groups.** The main explanatory variables were the initial timing of the audit and monitoring intervention by the ICT pharmacists (h) categorised as follows: <24 h, 24 to 72 h, 72 to 120 h, ≥120 h. [a]The odds ratios derived from the multivariable logistic regression were adjusted for age, sex, weight, eCCr, creatinine, AST, ALT, albumin, comorbidities (diabetes, cardiovascular diseases, renal failure, hypertension and dyslipidaemia), immunosuppressant use, ICU admission, vancomycin loading dose, concomitant drug use

(aminoglycoside, NSAIDs and piperacillin-tazobactam) and interventions of clinical pharmacists in charge of the wards. [b]The odds ratios derived from the multivariable logistic regression were adjusted for age, sex, eCCr, albumin, ICU admission, vancomycin loading dose and interventions of clinical pharmacists in charge of the wards. [c]The odds ratios derived from the multivariable logistic regression were adjusted for age, sex, comorbidities (diabetes, cancers, cardiovascular diseases, COPD, hepatic diseases, renal failure, hypertension, and dyslipidaemia), immunosuppressant use, ICU admission and interventions of clinical pharmacists in charge of the wards. [d]The odds ratios derived from the multivariable logistic regression were adjusted for age, sex, eCCr, albumin, vancomycin loading dose, concomitant drug use (piperacillin-tazobactam) and interventions of clinical pharmacists in charge of the wards. [e]The odds ratios derived from the multivariable logistic regression were adjusted for age, sex and eCCr. [f]The odds ratios derived from the multivariable logistic regression were adjusted for age, sex, comorbidities (diabetes, cancers, and cardiovascular diseases) and interventions of clinical pharmacists in charge of the wards. [g]The estimate from the multiple linear regression was adjusted for age, sex, weight, eCCr, creatinine, AST, ALT, albumin, comorbidities (diabetes, cardiovascular diseases), ICU admission, loading dose of vancomycin, concomitant drug use (aminoglycoside, NSAIDs and piperacillin-tazobactam), and interventions of clinical pharmacists in charge of the wards. [h]The odds ratios derived from the multivariable logistic regression were adjusted for age, sex, eCCr and interventions of clinical pharmacists in charge of the wards. [i]The odds ratios derived from the multivariable logistic regression were adjusted for age, sex, comorbidities (diabetes, cancers, cardiovascular diseases, COPD, hepatic diseases, renal failure, hypertension, and dyslipidaemia), immunosuppressant use and interventions of clinical pharmacists in charge of the wards. [j]The odds ratios derived from the multivariable logistic regression were adjusted for age, sex, weight, eCCr, creatinine, AST, ALT, albumin, comorbidities (diabetes and cardiovascular diseases), immunosuppressant use, ICU admission, vancomycin loading dose, concomitant drug use (aminoglycoside, NSAIDs and piperacillin-tazobactam) and interventions of clinical pharmacists in charge of the wards. [k]The odds ratios derived from the multivariable logistic regression were adjusted for age, sex, eCCr, ICU admission and interventions of clinical pharmacists in charge of the wards. ICT, infection control team; OR, odds ratio; 95% CI, 95% confidence interval; SD, standard deviation; SE, standard error; VCM, vancomycin; eCCr, estimate creatinine clearance; AST, aspartate aminotransferase, ALT, alanine aminotransferase; ICU, intensive care unit; NSAIDs, non-steroidal anti-inflammatory agents; COPD, chronic obstructive pulmonary disease; IQR, interquartile range.
(XLSX)

**S1 Fig. Forest plot of primary and secondary outcomes.** The main explanatory variables were the following continuous variables: the initial timing of the audit and monitoring intervention by the ICT pharmacists (h). **A).** (a) Maintenance of target trough concentration ranges for VCM during treatment duration, (b) Death within 30 days, (c) Implementation of de-escalation. Total number of patients was 638. No. of patients with events (%): (a) 307 (48.1), (b) 90 (14.1), (c) 206 (32.3). **B).** (a) Number of days of VCM administration. The average (SD) of number of days of VCM administration was 11.3 (7.4). (a) The odds ratios derived from the multivariable logistic regression or the estimates from the multiple linear regression were adjusted for age, sex, weight, eCCr, creatinine, AST, ALT, albumin, comorbidities (diabetes, cardiovascular diseases, renal failure, hypertension, and dyslipidaemia), immunosuppressant use, ICU admission, vancomycin loading dose, concomitant drug use (aminoglycoside, NSAIDs, piperacillin-tazobactam), and interventions of clinical pharmacists in charge of the wards. (b) The odds ratios derived from the multivariable logistic regression were adjusted for

age, sex, eCCr, albumin, comorbidities (cardiovascular diseases), ICU admission, the loading dose of vancomycin, and interventions of clinical pharmacists in charge of the wards. (c) The odds ratios derived from the multivariable logistic regression were adjusted for age, sex, comorbidities (diabetes, cancer, cardiovascular diseases, COPD, hepatic disease, renal failure, hypertension, and dyslipidaemia), immunosuppressant use, ICU admission, and interventions of clinical pharmacists in charge of the wards. ICT, infection control team; OR, odds ratio; 95% CI, 95% confidence interval; SD, standard deviation; SE, standard error; VCM, vancomycin; eCCr, estimate creatinine clearance; AST, aspartate aminotransferase, ALT, alanine aminotransferase; ICU, intensive care unit; NSAIDs, non-steroidal anti-inflammatory agents; COPD, chronic obstructive pulmonary disease.
(TIFF)

**S1 Checklist. STROBE statement—checklist of items that should be included in reports of observational studies.**
(DOCX)

## Acknowledgments

The authors would like to sincerely thank Editage (www.editage.jp) for English language editing.

## Author Contributions

**Conceptualization:** Hideki Sugita, Michiko Tanaka, Eisuke Inoue, Hironori Tanaka.

**Data curation:** Hideki Sugita, Natsumi Okada, Matoka Okamoto, Masakazu Abe, Masae Sekido, Michiko Tanaka.

**Formal analysis:** Hideki Sugita, Eisuke Inoue.

**Funding acquisition:** Hideki Sugita.

**Investigation:** Hideki Sugita, Natsumi Okada, Matoka Okamoto, Masakazu Abe, Masae Sekido, Michiko Tanaka.

**Methodology:** Hideki Sugita, Eisuke Inoue.

**Project administration:** Hideki Sugita.

**Resources:** Hideki Sugita, Natsumi Okada, Matoka Okamoto, Masakazu Abe, Masae Sekido, Michiko Tanaka.

**Software:** Hideki Sugita, Eisuke Inoue.

**Supervision:** Hideki Sugita, Eisuke Inoue, Hironori Tanaka.

**Validation:** Hideki Sugita, Eisuke Inoue.

**Visualization:** Hideki Sugita, Natsumi Okada, Matoka Okamoto, Masakazu Abe, Masae Sekido, Michiko Tanaka, Tatsuro Tamatukuri, Yuika Naito, Masayuki Yoshikawa, Eisuke Inoue, Hironori Tanaka.

**Writing – original draft:** Hideki Sugita, Eisuke Inoue.

**Writing – review & editing:** Hideki Sugita, Natsumi Okada, Matoka Okamoto, Masakazu Abe, Masae Sekido, Michiko Tanaka, Tatsuro Tamatukuri, Yuika Naito, Masayuki Yoshikawa, Eisuke Inoue, Hironori Tanaka.

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
