## [Decision Letter · Decision Letter 0]

4 Jun 2023

PONE-D-23-11831Evaluation of the initial timing of infection control pharmacist-driven audit and monitoring of vancomycin therapy in patients with infectious diseases: A retrospective observational studyPLOS ONE

Dear Dr. Sugita,

Thank you for submitting your manuscript to PLOS ONE. After careful consideration, we feel that it has merit but does not fully meet PLOS ONE’s publication criteria as it currently stands. Therefore, we invite you to submit a revised version of the manuscript that addresses the points raised during the review process.

We look forward to receiving your revised manuscript.

Kind regards,

Keiko Hosohata, Ph.D.

Academic Editor

PLOS ONE

“I have read the journal's policy and the authors of this manuscript have the following competing interests: EI received lecture fees from Pfizer Japan Inc., Bristol-Myers Squibb Co., and Eisai Co., Ltd., and consulting fees from Nippontect Systems Co., Ltd., and Cyberdine, Inc. The other authors declare no conflicts of interest.”

Reviewers' comments:

Reviewer's Responses to Questions

**Comments to the Author**

1. Is the manuscript technically sound, and do the data support the conclusions?

Reviewer #1: Partly

Reviewer #2: No

Reviewer #3: Partly

Reviewer #4: Yes

Reviewer #5: Partly

2. Has the statistical analysis been performed appropriately and rigorously? 

Reviewer #1: Yes

Reviewer #2: Yes

Reviewer #3: Yes

Reviewer #4: Yes

Reviewer #5: Yes

3. Have the authors made all data underlying the findings in their manuscript fully available?

Reviewer #1: Yes

Reviewer #2: Yes

Reviewer #3: Yes

Reviewer #4: Yes

Reviewer #5: Yes

4. Is the manuscript presented in an intelligible fashion and written in standard English?

Reviewer #1: No

Reviewer #2: No

Reviewer #3: No

Reviewer #4: Yes

Reviewer #5: Yes

5. Review Comments to the Author

Reviewer #1: The authors have asked an important question around timing of audits for Antimicrobial stewardship reviews. As pharmacist resources are limited, it is important to conduct the reviews when their will be optimal benefit. The authors however chose to focus as their primary outcome on duration of appropriate vancomycin trough which was unaffected by timing of the audits (not entirely surprising). Their secondary endpoint of days on vancomycin therapy is really the more important outcome. While the authors state the duration of therapy is longer with increase time from prescription to audit. The data would suggest it is shortest in those patients who are audited between 24 and 72 hours (the second group) although the table is a bit difficult to interpret (table 3). It is also not clear exactly what the difference is between these groups in days.

Additionally there are a number of grammatical issues that make the manuscript difficult to follow. I think the question is interesting but it needs major revisions to make the results more understandable.

Detailed review:

Line 65: require should be pleural - requires

Line 66 Not only by single professionals does not make sense - would suggest rewording or taking this out

Line 72: treatment should pleural

Line 74: there should be no comma after proactive monitoring

Line 119: hypocytosis is not correct - it should be cytopenia

Line 125: and should be or

Line 129: the dates outlines are confusing - initially you suggest including patients from Jan 12019-May 31, 2021 but then say pharmacists collected information from March 1, 2019 to April 7 2021 and then include patients from June 28 2021 to Feb 24 2023. Please make this paragraph clearer

Line 137: continued achievement does not really make sense - it should be duration of optimal blood vanco concentration

Line 153: grammar is off in this sentence - not sure what in one to three per week means - please revise

Line 162: I think this should say data were obtained including not for. Also presumably you collected data on trough level but never state when the trough level was collected with respect to the first dose - was it the same for all patients - this is important to explain

Line 167: Again - this is not clear - I think you are trying to say Reasons for vancomycin cessation were classified as ...

Line 177: I do not think efficacy is the right term here - this sentence does not make sense - you are trying to determine if there is a difference in optimal trough vancomycin level depending on timing of audit adjusted for covariables.

Line 196: sample size calculation is also not clear - you have stated you need 614 in both groups but there are 4 different time periods that you have compared for auditing so you need to specify how many patients you need in each of these auditing periods to detect a difference in duration of optimal trough levels.

Line 233: whereas does not make sense

Line 255: again - this needs to be reworded - I think you are trying to say common adverse events caused by vancomycin occurred in 44 patients including renal failure and skin rash.

Line 262: this section needs to be a bit clearer - authors should state how many or what proportion of patients achieved target trough levels for what duration in each audit time period. Table 3 suggests that a small number of patients achieved target concentrations but it is not clear if this is at the time of the audit or what the duration of time it. This table needs to be reworked to make it clearer

Page 21 table 3 - similarly I am not clear on the number of days of vancomycin administration - I think it would make more sense to have the average per person in each of the audit time periods. I assume in the table the authors have the total number of days which is not really relevant.

Page 25: subgroup analysis table also is confusing and should have the average number of days of vanomycin administration in each of these settings.

Line 358: common after prescribing diagnosis

Line 360: by ICT pharmacists and the maintenance

Line 362-363: needs to be reworded - unclear

Line 375: should read clinical pharmacists on the wards

Line 383: this needs to have some context - it would make sense to say that only 48.1% of all patients achieved target trough (if I am reading the study correctly) and then you could hypothesize that it was due to not being treated with an initial loading dose etc.

Reviewer #2: Dear authors,

Thank you for bringing up an important issue. I acknowledge the welcoming theme of the manuscript, however, I agree that a longitudinal study incorporating real-time plasma drug concentration analysis would provide a more comprehensive reflection of the problem. Prospective study approach would enable the researcher a deeper insight, generate robust evidence, and incorporate the findings into the healthcare setups more effectively.

After carefully reviewing the manuscript, I regret to inform you that I have reached the conclusion to decline its publication based on the following points:

1. Language: Although the grammar is technically correct, the writing lacks academic clarity, robustness, and maybe precision.

2. Problem statement: The authors struggled to effectively articulate and coherently explain the main problem, resulting in ineffective and substandard writing.

3. Reproducibility: The manuscript exhibits a lack of coherence and clarity in its writing, making it challenging to reproduce the findings effectively.

4. Headings: Several subheadings within the methods and discussion sections are inappropriate and improperly ordered, deviating from the standards expected in reputable journals such as PLOS ONE.

5. Figures: It is essential for standard research articles to include at least one significant figure that succinctly presents the results to readers. Unfortunately, the manuscript only includes a single figure depicting exclusion and inclusion criteria, lacking any prominent visuals.

Overall: Based on the journal metrics, I believe this manuscript falls short of meeting the required standards for publication in the PLOS ONE.

Reviewer #3: Authors conducted the retrospective observational study to presents the impact of the initial timing of audit and monitoring of vancomycin by Infection Control Team (ICT) pharmacists, on maintenance of the target vancomycin trough concentration in a university hospital in Tokyo, Japan from 1 Jan 2019 to 31 May 2021. Results of present study suggest that early initiation of a comprehensive audit and monitoring by ICT pharmacists did not affect the maintenance of the target vancomycin blood concentration, but reduced the duration of vancomycin administration. The authors offer an interesting empirical analysis of the impact of ICT pharmacists on vancomycin administration. The methods used are reasonable, as they prospective enrolled eligible patients (n=638) with an infection or suspected infection. Authors applied suitable statistical model (multivariable logistic regression and multivariate linear regression) to test the effectiveness of the initial timing of the intervention by ICT pharmacists. The results of these analyses are clearly presented and discussed. The main limitation of this study lies in the observational design as observational studies are unable to perfectly control for all biases. Furthermore, the study is quite interesting but it has many limitations with poor write-up. The authors should consider following changes to improve the manuscript.

Major Suggestions:

Comment 1: Abstract section: Background needs reconsideration in relevant write-up. Authors should brief the information related to the significance of the study instead of writing aims of the study. In methodology section (Line 36-37), it would be better to remove outcome of the study as it is misfit in the context of methodology. Line 102-103 needs correction grammatically. First letter of Eastern should be capital.

Comment 2: The introduction could be improved by providing a more thorough overview of the current literature on the early monitoring and feedback of infectious disease treatment and expanding on the strategies that can be used to address antimicrobial resistance. Finally, it would be helpful to provide a more detailed description of the research question and objectives for the study.

Comment 3: The methodology of this study is somewhat well-explained and comprehensive. However, certain factors should be considered in order to further improve the quality of the study. First, the reliability of the collected data must be considered. It is important to ensure that data collected from the medical records are consistent with the collected data from the practical ICT database. Additionally, sufficient details must be provided in order to prove that ethical standards were followed during the study. For example, additional information must be provided on how rights and welfare of the participants were protected throughout the whole study duration. Finally, the explanation of the statistical analysis used to assess the collected data must be further elaborated to prove the validity of the study’s conclusions.

Comment 4: Authors should review manuscript grammatically and repetition of sentences throughout the manuscript should be discouraged. It is suggested to improve the sentence structures throughout the manuscript as well.

Comment 5: Authors should describe the concentration related adverse effects of the Vancomycin particularly in organ compromised patients (kidney/liver disease).

Comment 6: Heading of sample size should be in discussed in methodology section instead of after statistical analyses. Discussion heading is enough, no need to write subheadings as key results and interpretations. Discussion section is poorly written and must be improved by addition of logical justification of the results. In addition; the safety profile of the drug (Vancomycin) should also be discussed in discussion section

Comment 7: Conclusion needs improvement and should be outcome based; sentence related to safety of interventions in conclusion section is inappropriate.

Comment 8: Authors should add the Future perspective of the study. Manuscripts lack the implications of findings of the study. It would be more appropriate if authors add its clinical significance.

Decision: Manuscript requires major revision.

Reviewer #4: Hideki Sugita, et al., carried out a retrospective observational study to investigate the initial timing of infection control pharmacist-driven audit and monitoring of vancomycin therapy in patients with infectious diseases.

To the best of my knowledge, the study has substantial scientific merit and reads quite well. The experiments, and statistical analyses performed were according to standard and were described in sufficient detail and meets all applicable standards for the ethics of experimentation and research integrity.

The limitations of the study were also highlighted.

There however some minor grammatical/typographical errors.

I will suggest the manuscript is thoroughly revised to correct the grammatical/typographical errors.

Reviewer #5: The manuscript presents a retrospective observational study that evaluates the initial timing of infection control pharmacist-driven audit and monitoring of vancomycin therapy in patients with infectious diseases

Flowchart representing the patient selection process should be provided

Clinical implications of the study's findings and how they can contribute to improving patient outcomes should be discussed

6. PLOS authors have the option to publish the peer review history of their article (what does this mean?). If published, this will include your full peer review and any attached files.

Reviewer #1: No

Reviewer #2: No

Reviewer #3: **Yes: **Mehmood Ahmad

Reviewer #4: **Yes: **Isaac Akefe Oluwatobi

Reviewer #5: No

---

## [Author Response · Author response to Decision Letter 0]

8 Aug 2023

Response to Editor’s and Reviewer’s comments

We sincerely appreciate your valuable comments and have endeavoured to address them all. Our point-by-point responses are provided below:

Response: Thank you for your indication. We have carefully rechecked the manuscript according to the PLOS ONE style.

“I have read the journal's policy and the authors of this manuscript have the following competing interests: EI received lecture fees from Pfizer Japan Inc., Bristol-Myers Squibb Co., and Eisai Co., Ltd., and consulting fees from Nippontect Systems Co., Ltd., and Cyberdine, Inc. The other authors declare no conflicts of interest.”

Response: We added our updated Competing Interests statement in our cover letter as follows: 

"This does not alter our adherence to PLOS ONE policies on sharing data and materials."

Response: Thank you for your indication. When we first submitted the dataset to PLOS ONE, we had removed some of the data in consideration of personal information at our investigators' discretion. As the Personal Information Protection Law was revised in April 2023 in Japan, we conferred with our ethics committee regarding the disclosure of the dataset of this study. They informed us that the dataset could not be disclosed for the protection of personal information. In accordance with the recommendations from the Ethics Committee, we have made the following changes. We apologise that we are unable to comply with PLOS ONE policy.

Data availability:

“Concerning data availability, there are restrictions on disclosing individual patients’ information for ethical considerations based the recommendation of the Ethics Committee on Research Involving Human Subjects at Showa University. Due to the limited number of patients admitted to our hospital and the small population size at our research centre, only the option of pseudo-anonymization of personal information is available to us. As such, the data set cannot be made available.” (p.34-35, line529-535).

Reviewers' comments: Reviewer's Responses to Questions

Comments to the Author€b01. Is the manuscript technically sound, and do the data support the conclusions? The manuscript must describe a technically sound piece of scientific research with data that supports the conclusions. Experiments must have been conducted rigorously, with appropriate controls, replication, and sample sizes. The conclusions must be drawn appropriately based on the data presented. 

Reviewer #1: Partly

Reviewer #2: No

Reviewer #3: Partly

Reviewer #4: Yes

Reviewer #5: Partly

Response: Thank you for your comments. We have carefully reviewed the comments and have revised the manuscript accordingly. 

2. Has the statistical analysis been performed appropriately and rigorously? 

Reviewer #1: Yes

Reviewer #2: Yes

Reviewer #3: Yes

Reviewer #4: Yes

Reviewer #5: Yes

Response: Thank you for the feedback.

3. Have the authors made all data underlying the findings in their manuscript fully available? The PLOS Data policy requires authors to make all data underlying the findings described in their manuscript fully available without restriction, with rare exception (please refer to the Data Availability Statement in the manuscript PDF file). The data should be provided as part of the manuscript or its supporting information, or deposited to a public repository. For example, in addition to summary statistics, the data points behind means, medians and variance measures should be available. If there are restrictions on publicly sharing data—e.g. participant privacy or use of data from a third party—those must be specified.

Reviewer #1: Yes

Reviewer #2: Yes

Reviewer #3: Yes

Reviewer #4: Yes

Reviewer #5: Yes

Response: Thank you. We made some corrections in accordance with what is pointed out in "Journal requirements: 3" above.

4. Is the manuscript presented in an intelligible fashion and written in standard English? PLOS ONE does not copyedit accepted manuscripts, so the language in submitted articles must be clear, correct, and unambiguous. Any typographical or grammatical errors should be corrected at revision, so please note any specific errors here.

Reviewer #1: No

Reviewer #2: No

Reviewer #3: No

Reviewer #4: Yes

Reviewer #5: Yes

Response: Thank you for your comments. The text has been carefully and meticulously edited again for language throughout the manuscript, including the areas you pointed out.

5. Review Comments to the Author

Reviewer #1: The authors have asked an important question around timing of audits for Antimicrobial stewardship reviews. As pharmacist resources are limited, it is important to conduct the reviews when their will be optimal benefit. The authors however chose to focus as their primary outcome on duration of appropriate vancomycin trough which was unaffected by timing of the audits (not entirely surprising). Their secondary endpoint of days on vancomycin therapy is really the more important outcome. While the authors state the duration of therapy is longer with increase time from prescription to audit. The data would suggest it is shortest in those patients who are audited between 24 and 72 hours (the second group) although the table is a bit difficult to interpret (table 3). It is also not clear exactly what the difference is between these groups in days. Additionally there are a number of grammatical issues that make the manuscript difficult to follow. I think the question is interesting but it needs major revisions to make the results more understandable. 

Response:

Thank you for pointing this out. The main analysis of this study was to examine the explanatory variables as continuous variables. For categorised variables, we added the explanatory document shown below to understand easily as exploratory categorisation. To prevent any confusion in the interpretation of Table 3, we have moved the four categories of exploratory categorisation to the supporting information section (S3 Table). 

“The initial timing of the audit and monitoring intervention by ICT pharmacists was obtained as continuous variables from our ICT pharmacists’ practical database. Pharmacist-led ASP interventions have been implemented once or three times a week in previous studies and have established clinical benefits such as reduced antimicrobial use and shorter treatment times [11, 19]. In our hospital, pharmacist-led patient monitoring of anti-MRSA drug use was conducted once or twice a week during this study period. In this study, the timing of intervention initiation was exploratively categorised into four groups, considering that the intervention timings in previous studies would take place at least once to three times a week as follows: <24 hours, 24 to 72 hours, 72 to 120 hours, ≥120 hours [11, 19].” (p.9-10, line 181-194).

Detailed review: 

Response: Thank you for your valuable feedback. The text has been revised accordingly.

Line 65: require should be pleural – requires

Response: We have revised the term to “requires”

“Infectious disease treatment, prevention and control requires a wide range of educational activities by medical doctors, nurses, pharmacists, and other medical providers.” (p.4, line 69-70).

Line 66 Not only by single professionals does not make sense - would suggest rewording or taking this out

Response: We have excluded the following in the revised manuscript: 

", not only by single professionals but also" (p.4, line 70).

Line 72: treatment should pleural

Response: We revised as follows: 

“Early monitoring and feedback on the treatment of infectious diseases are some of the methods for optimising antimicrobial treatment throughout the treatment period.” (p.4-5, line 76-78).

Line 74: there should be no comma after proactive monitoring

Response: We revised with other reviewers’ comments as follows: 

“These also include proactive monitoring, administering antimicrobial agents, reviewing infectious disease test results, and providing necessary feedback [7].” (p.5, line 78-80).

Line 119: hypocytosis is not correct - it should be cytopenia

Response: We have revised the term to “cytopenia” (p.8, line 137).

Line 125: and should be or

Response: We revised “and” to “or” (p.8, line 143).

Line 129: the dates outlines are confusing - initially you suggest including patients from Jan 1 2019-May 31, 2021 but then say pharmacists collected information from March 1, 2019 to April 7 2021 and then include patients from June 28 2021 to Feb 24 2023. Please make this paragraph clearer

Response: We deleted duplicate dates listed in "Study design and settings” section and clarified the contribution: 

“ICT pharmacists accumulated patient information from 1st March 2019 to 7th April 2021 as a general ICT practice at that time. For this study, investigators retrospectively collected data at baseline from our ICT pharmacists’ practical database and electronic medical records from 28th June 2021 to 24th February 2023.” (p.8, line 147-150).

Line 137: continued achievement does not really make sense - it should be duration of optimal blood vanco concentration

Response: We have made the following revision: 

“The primary outcome was the maintenance of target vancomycin trough blood concentration during treatment, whose range was defined from 10 to 20 μg/ml as the trough level.” (p.9, line 157-159).

Line 153: grammar is off in this sentence - not sure what in one to three per week means - please revise

Response: We deleted and amended the document to read as follows:

“In this study, the timing of intervention initiation was exploratively categorised into four groups, considering that the intervention timings in previous studies would take place at least once to three times a week as follows: <24 hours, 24 to 72 hours, 72 to 120 hours, ≥120 hours [11, 19].” (p.10, line 173-176).

Line 162: I think this should say data were obtained including not for. Also presumably you collected data on trough level but never state when the trough level was collected with respect to the first dose - was it the same for all patients - this is important to explain

Response: Thank you for your comments. At this facility, in some cases, a non-infection specialised-physician initiated vancomycin administration and performed the concentration measurement order, while in other cases, clinical pharmacists collaborated with general physicians on the design of the first vancomycin administration, including requests from the ICT pharmacist. Therefore, the timing of vancomycin concentration measurements and collection may not have been consistent with the trough levels. 

We revised some modifications to the method section with other reviewers’ comments, and added documentation on the accuracy of concentration measurements and blood sampling in the discussion section.

Materials and Methods: Data collection and variables

“We retrospectively collected the following data from our ICT pharmacists’ practical database and electronic medical records when necessary: 1) the time of the audit and monitoring intervention by ICT pharmacists; diagnosis department; interventions of clinical pharmacists in charge of the wards; vancomycin-related data (duration of administration and blood trough concentration), 2) age; sex; height; weight; comorbidities; immunosuppressant use; intensive care unit (ICU) admission; white blood cells, neutrophils, platelets, creatinine and blood urea nitrogen levels; estimated glomerular filtration rate; aspartate aminotransferase, alanine aminotransferase, γ-glutamyl transpeptidase and albumin levels; concomitant drug use; other vancomycin-related data (the prescribed date and time, initial dose and dosage); data on diagnoses; suspected infections; the causative or detected organisms. In some cases, general physicians initiated vancomycin administration and performed the concentration measurement order, while in other cases, clinical pharmacists collaborated with general physicians who were involved in the design of the first vancomycin administration strategy, including requests from the ICT pharmacist. Therefore, the timing of the first vancomycin concentration measurement was not consistent for each patient.” (p.10-11, line 176-191).

Discussion:

“During the study period, only 43.1% of the clinical pharmacists were involved in designing the first dose of vancomycin and suggesting the timing of trough concentration measurements, before prescription of vancomycin by the physicians (Table 1). There may have been a lack of adequate dosing and monitoring from the start of vancomycin treatment, although multivariate analysis was performed by adjusting the implementation of initial intervention by clinical pharmacists. There may be a need to strengthen and consider an additional system of collaboration between clinical pharmacists and ICT pharmacists.” (p30, line 416-422).

Line 167: Again - this is not clear - I think you are trying to say Reasons for vancomycin cessation were classified as ...

Response: Thank you for your feedback. We have revised as follows:

“Reasons for vancomycin cessation were classified as cure or improvement, death …...” (p.11, line 197).

Line 177: I do not think efficacy is the right term here - this sentence does not make sense - you are trying to determine if there is a difference in optimal trough vancomycin level depending on timing of audit adjusted for covariables.

Response: We agree your indication. We revised to “effectiveness” as follows:

“Multivariable logistic regression analysis for the primary outcome was performed to test the effectiveness of the periods between the prescription order for vancomycin and the initial audit and monitoring period by the pharmacist in charge of the ICT,…..” (p.12, line 220-222).

Line 196: sample size calculation is also not clear - you have stated you need 614 in both groups but there are 4 different time periods that you have compared for auditing so you need to specify how many patients you need in each of these auditing periods to detect a difference in duration of optimal trough levels.

Response: 

Thank you for bringing this to our attention. As the main explanatory variable was a continuous variable in this study, we have revised it as follows so that it can be calculated clearly:

Sample size

“Furthermore, considering that all cases were implemented in audit and monitoring practices, there was an estimated odds ratio of 2.0 in 2 SD increase in the period (h) between the prescription order for vancomycin and the initial audit and monitoring by the ICT pharmacist, and a correlation coefficient with covariates of 0.6.” (p11-12, line 208-212).

Line 233: whereas does not make sense

Response: We revised as follows: 

“Comorbidities included cancer (56.6%), cardiovascular disease (46.6%), and renal failure (20.2%).” (p.15, line 283-284).

Line 255: again - this needs to be reworded - I think you are trying to say common adverse events caused by vancomycin occurred in 44 patients including renal failure and skin rash.

Response: We revised as follows: 

“Common adverse events caused by vancomycin occurred in 72 patients and included those with renal impairment (44, 6.9%) and skin rash or allergy (28, 4.7%).” (p.20, line 306-308).

Line 262: this section needs to be a bit clearer - authors should state how many or what proportion of patients achieved target trough levels for what duration in each audit time period. 

Response: Thank you for your comments. We have added the following sentence:

“Maintenance of target vancomycin trough blood concentration during treatment was achieved in 307 (48.1%) patients in each patients' treatment period (Table 3). The four exploratory classification groups were as follows: <24 h (67, 10.5%), ≥24 h, <72 h (137, 21.5%), ≥72 h, <120 h (64, 10%), and ≥120 (39, 6.1%) (S3 Table).” (p.22, line 314-317).

Table 3 suggests that a small number of patients achieved target concentrations but it is not clear if this is at the time of the audit or what the duration of time it. This table needs to be reworked to make it clearer 

Response: We revised Table 3 along with the other tables to demonstrate if the vancomycin target concentration was maintained during the treatment period.

“Maintenance of target VCM trough blood concentration during treatment” (Table 3, S3 Table, S4 Table, S5 Table).

Page 21 table 3 - similarly I am not clear on the number of days of vancomycin administration - I think it would make more sense to have the average per person in each of the audit time periods. I assume in the table the authors have the total number of days which is not really relevant.

Response: Thank you for your suggestions. We have included the information in Table 3 for facilitating the interpretation of results for continuous explanatory variables. The results of the analysis of the categorical variables, which were categorised into four exploratory groups, have been moved to the supporting information (S3 Table). With respect to the number of days of vancomycin administration, the average number of days of administration when divided into the four groups was also included. In accordance with the comments from Reviewer 2 comments, we have added the "S1 Fig" as supporting information (p.47-48, line820-845).

Page 25: subgroup analysis table also is confusing and should have the average number of days of vancomycin administration in each of these settings.

Response: Thank you for your suggestions and we have accordingly revised the text. The results of the analysis of the categorical variables, which were categorised into four exploratory groups, have been moved to the supporting information (S5 Table). With respect to the number of days of vancomycin administration, the average number of days of administration when divided into the four groups, was also included.

Line 358: common after prescribing diagnosis

Response: We revised as follows: 

“Despite differences in patient populations, such as different departments commonly making the diagnoses and critical care settings, there was no significant correlation between the timing of the audit and the initiation of monitoring by ICT pharmacists and the maintenance of target vancomycin trough blood concentrations.” (p.29, line 392-396).

Line 360: by ICT pharmacists and the maintenance

Response: The following revision has been made: 

“by ICT pharmacists and the maintenance of target vancomycin trough blood concentrations.” (p.29, line 395-396).

Line 362-363: needs to be reworded – unclear

Response: We have revised accordingly: 

“There was a significant association between the number of days of vancomycin administration and the timing of the audit and monitoring initiation by the ICT pharmacists.” (p.29, line 396-398).

Line 375: should read clinical pharmacists on the wards

Response: We revised as follows: 

“clinical pharmacists associated with the wards….” (p29, line 408-409).

Line 383: this needs to have some context - it would make sense to say that only 48.1% of all patients achieved target trough (if I am reading the study correctly) and then you could hypothesize that it was due to not being treated with an initial loading dose etc.

Response: Thank you for your comments. We have made the following revisions:

“Overall, 31.7% of the patients were treated with an initial vancomycin loading dose. In the guidelines proposed during the study period, the loading dose might be considered, in order to rapidly achieve the target concentrations in seriously ill patients [15, 16]. We may not have been able to appropriately evaluate whether the patient was critically ill or not, since we had not assessed the severity of the disease in each patient in our general practice in this investigation. Only 48.1% of all the patients maintained their target vancomycin trough blood concentration during therapy. The loading dose may not have been administered to all critically ill patients as recommended by the guidelines, preventing them from reaching the initial target concentration.” (p.30, line 423-431).

Reviewer #2: Dear authors, Thank you for bringing up an important issue. I acknowledge the welcoming theme of the manuscript, however, I agree that a longitudinal study incorporating real-time plasma drug concentration analysis would provide a more comprehensive reflection of the problem. Prospective study approach would enable the researcher a deeper insight, generate robust evidence, and incorporate the findings into the healthcare setups more effectively. After carefully reviewing the manuscript, I regret to inform you that I have reached the conclusion to decline its publication based on the following 

points:1. Language: Although the grammar is technically correct, the writing lacks academic clarity, robustness, and maybe precision.

Response: Thank you for your comments. The text has been revised extensively for grammatical correctness and language. 

2. Problem statement: The authors struggled to effectively articulate and coherently explain the main problem, resulting in ineffective and substandard writing.

Response: Thank you for your comments. We have taken the necessary steps to ensure that your concerns are addressed. The manuscript has been revised for language, grammar and ensured that the writing meets the standards of your esteemed journal.

3. Reproducibility: The manuscript exhibits a lack of coherence and clarity in its writing, making it challenging to reproduce the findings effectively.

Response: Thank you for your valuable feedback. We reviewed the methods section in its entirety and have revised it accordingly.

4. Headings: Several subheadings within the methods and discussion sections are inappropriate and improperly ordered, deviating from the standards expected in reputable journals such as PLOS ONE.

Response: We revised the hierarchical representation of the subheadings by referring to the PLOS ONE submission guideline and other previous studies of PLOS ONE.

5. Figures: It is essential for standard research articles to include at least one significant figure that succinctly presents the results to readers. Unfortunately, the manuscript only includes a single figure depicting exclusion and inclusion criteria, lacking any prominent visuals. 

Response: Thank you for bringing this to our attention. Except for one secondary outcome by multiple regression analysis, we believed it would be less impressive to show the primary and secondary outcomes in figures using odds ratios of Table 3. As a result, we used Table 3 to incorporate them into the text. We made it into a figure in the forest plot as supporting information (S1 Fig, p.47-48, line820-845).

Overall: Based on the journal metrics, I believe this manuscript falls short of meeting the required standards for publication in the PLOS ONE.

Reviewer #3: Authors conducted the retrospective observational study to presents the impact of the initial timing of audit and monitoring of vancomycin by Infection Control Team (ICT) pharmacists, on maintenance of the target vancomycin trough concentration in a university hospital in Tokyo, Japan from 1 Jan 2019 to 31 May 2021. Results of present study suggest that early initiation of a comprehensive audit and monitoring by ICT pharmacists did not affect the maintenance of the target vancomycin blood concentration, but reduced the duration of vancomycin administration. The authors offer an interesting empirical analysis of the impact of ICT pharmacists on vancomycin administration. The methods used are reasonable, as they prospective enrolled eligible patients (n=638) with an infection or suspected infection. Authors applied suitable statistical model (multivariable logistic regression and multivariate linear regression) to test the effectiveness of the initial timing of the intervention by ICT pharmacists. The results of these analyses are clearly presented and discussed. The main limitation of this study lies in the observational design as observational studies are unable to perfectly control for all biases. Furthermore, the study is quite interesting but it has many limitations with poor write-up. The authors should consider following changes to improve the manuscript.

Major Suggestions:

Comment 1: Abstract section: Background needs reconsideration in relevant write-up. Authors should brief the information related to the significance of the study instead of writing aims of the study. In methodology section (Line 36-37), it would be better to remove outcome of the study as it is misfit in the context of methodology. Line 102-103 needs correction grammatically. First letter of Eastern should be capital.

Response: Thank you for your comments. We reviewed each of your points and revised the Abstract section as follows:

[Background] 

“Early monitoring and feedback on the treatment of infectious diseases are some of the methods for optimising antimicrobial treatment throughout the treatment period. Prospective audits and feedback interventions have also been shown to improve antimicrobial use and reduce antimicrobial resistance. We examined the appropriate use of antimicrobials by focusing on the initial timing for audits and feedback intervention of antimicrobial prescription by Infection Control Team pharmacists.” (p.2, line 29-34).

[Methods] 

We meant the definition of primary outcome. We have made amends to clarify that it is a description of the definition rather than a description of the primary outcome as follows:

“The definition of primary outcome was the maintenance of target vancomycin trough blood concentrations of 10–20 μg/ml during treatment.” (p.3, line 39-40).

[Line 102-103]

We revised this sentence below:

“This retrospective, observational study was conducted at an 815-bed university hospital in Eastern Tokyo, Japan.” (p.7, line119-120).

Comment 2: The introduction could be improved by providing a more thorough overview of the current literature on the early monitoring and feedback of infectious disease treatment and expanding on the strategies that can be used to address antimicrobial resistance. Finally, it would be helpful to provide a more detailed description of the research question and objectives for the study.

[Comment 2]

Response: Thank you for your comments. We have underlined the changes made and revised the Introduction section as follows:

“Twenty years ago, it was understood that avoiding overuse and inappropriate use of antimicrobials can reduce the development of bacterial resistance [6]. Early monitoring and feedback on the treatment of infectious diseases are some of the methods for optimising antimicrobial treatment throughout the treatment period. These also include proactive monitoring, administering antimicrobial agents, reviewing infectious disease test results, and providing necessary feedback [7]. Prospective audits and feedback (PAF) interventions also improve antimicrobial use, reduce antimicrobial resistance, and lower Clostridioides difficile infection rates without adversely affecting patient outcomes [8-11]. The following are reported as the timing of interventions: at the time it was deemed necessary to alter the choice of an antimicrobial agent or dosage; when the results of infectious disease tests are known; when the effectiveness of treatment is determined; when the route of administration is changed; and when the drug is administered for a long duration [7]. However, direct intervention and feedback by the Infection Control Team (ICT) and Antimicrobial Stewardship Team (AST), whose core members are infectious disease specialists and clinical pharmacists, are limited in their ability to reach all patients because of time and human resource constraints. The implementation of appropriate antimicrobial use differs between institutions with dedicated ASPs and those without [12]. The timing of AST interventions, which could be daily, thrice a week, or once a week, is tailored to the feasibility of each facility and contributes to the appropriate use of antimicrobials without adversely affecting patient outcomes [8, 10, 11]. Weekly AST interventions reduced antimicrobial use, long-term use rates, drug resistance rates among Pseudomonas aeruginosa and MRSA infections, and costs of specific antimicrobials, when compared to those of interventions after a certain period of antimicrobial use (i.e., more than 14 days) [11]. In a clinical pharmacist-led intervention, 70% of the dose and antimicrobial changes were reported to have been accepted following pharmacist-led monitoring and feedback on infectious disease treatment (AST rounds of 1 h each, thrice a week), with no specialist training in infectious diseases [13]. In contrast, a dilemma has been identified with the economic benefits of antimicrobial stewardship (AS) activities, including interactive educational interventions in hospital management, whereby infectious disease specialists spent a considerable amount of time improving clinical effectiveness, i.e., reducing mortality rates and length of hospital stay [8]. A robust ASP required 1.0 full-time pharmacist and 0.25 full-time physician per 100 beds [14]. Therefore, considering the huge labour burden and costs of increasing the number of reviews and PAFs as a practice of ASTs and ICTs, it is important to consider reasonable options with a multi-professional division of labour that utilises expertise from across the board. Among these issues, no reports have examined the appropriate use of antimicrobial drugs with an emphasis on the initial timing to audit and feedback by ICT pharmacists who perform antimicrobial drug logistical audits.

This study assessed the impact of the initial timing of the audit and monitoring intervention of vancomycin by ICT pharmacists, in patients who had developed or were suspected of having an infection, on the maintenance of target vancomycin trough blood concentration during treatment.” (p.4-7, line 75-115).

Additional references:

6. Bantar C, Sartori B, Vesco E, Heft C, Saul M, Salamone F, et al. A hospitalwide intervention program to optimize the quality of antibiotic use: impact on prescribing practice, antibiotic consumption, cost savings, and bacterial resistance. Clin Infect Dis. 2003;37: 180-186. https://doi.org/10.1086/375818.

14. Pulcini C, Binda F, Lamkang AS, Trett A, Charani E, Goff DA, et al. Developing core elements and checklist items for global hospital antimicrobial stewardship programmes: a consensus approach. Clin Microbiol Infect. 2019;25: 20-25. https://doi.org/10.1016/j.cmi.2018.03.033.

Comment 3: The methodology of this study is somewhat well-explained and comprehensive. However, certain factors should be considered in order to further improve the quality of the study. First, the reliability of the collected data must be considered. It is important to ensure that data collected from the medical records are consistent with the collected data from the practical ICT database.

Response: Thank you for your comments.

[Comment 3-1] 

We added a description of the position of the ICT practical database in "Materials and Methods; Study population" section.

“The daily ICT practical implementation record in the pharmacy department comprises the ICT practical database. The patient information included in this database was extracted from the electronic medical record. Based on the ICT practical database, the investigators acquired the necessary variable data from each patient's electronic medical records to conduct this investigation.” (p.8, line 150-154).

In addition, we revised the data collection information (underlined sentences 1), 2)) separately for the ICT practical database and the electronic medical record in "Materials and Methods; Data collection and variables ".

“We retrospectively collected the following data from our ICT pharmacists’ practical database and electronic medical records when necessary: 1) the time of the audit and monitoring intervention by ICT pharmacists; diagnosis department; interventions of clinical pharmacists in charge of the wards; vancomycin-related data (duration of administration and blood trough concentration), 2) age; sex; height; weight; comorbidities; immunosuppressant use; intensive care unit (ICU) admission; white blood cells, neutrophils, platelets, creatinine and blood urea nitrogen levels; estimated glomerular filtration rate; aspartate aminotransferase, alanine aminotransferase, γ-glutamyl transpeptidase and albumin levels; concomitant drug use; other vancomycin-related data (the prescribed date and time, initial dose and dosage); data on diagnoses; suspected infections; the causative or detected organisms. ” (p.10, line 176-186).

Furthermore, we added information regarding the investigators at the end of the 'Data collection and variables' section as follows:

“Three trained ICT pharmacists (HS, MT, and YN) implemented data collection to construct the ICT practical database. In addition, five investigators (HS, NO, MO, KA, and MS) conducted data collection for this study from electronic medical records based on standardised work procedures.” (p11, line 192-196).

Additionally, sufficient details must be provided in order to prove that ethical standards were followed during the study. For example, additional information must be provided on how rights and welfare of the participants were protected throughout the whole study duration. 

[Comment 3-2]

Response: Thank you for your indication. We have revised “Ethical approval statement” section as follows:

“The study was conducted in compliance with the Declaration of Helsinki for protecting the rights and welfare of the patients and followed the Strengthening the Reporting of Observational studies in Epidemiology (STROBE) guidelines for reporting [22].” (p.14-15, line 264-267).

Additional reference)

22. von Elm E, Altman DG, Egger M, Pocock SJ, Gotzsche PC, Vandenbroucke JP, et al. The Strengthening the Reporting of Observational Studies in Epidemiology (STROBE) statement: guidelines for reporting observational studies. PLoS Med. 2007;4: e296. https://doi.org/10.1371/journal.pmed.0040296.

Finally, the explanation of the statistical analysis used to assess the collected data must be further elaborated to prove the validity of the study’s conclusions.

[Comment 3-3]

Response: Thank you for your comment. We revised as the following to "Statistical analysis” section, and added the underlined sentences:

“Multivariable logistic regression analysis for the primary outcome was performed to test the effectiveness of the periods between the prescription order for vancomycin and the initial audit and monitoring period by the pharmacist in charge of the ICT, adjusted by age, sex, weight, eCCr, creatinine, AST, ALT, albumin, comorbidities (diabetes, cardiovascular diseases, renal failure, hypertension, and dyslipidaemia), immunosuppressant use, ICU admission, vancomycin loading dose, concomitant drug use (aminoglycoside, NSAIDs, piperacillin-tazobactam), and interventions of clinical pharmacists in charge of the wards, at baseline. Confounders that might alter the pharmacokinetics and pharmacodynamics of vancomycin and that might influence the environment in which treatment is received were used to adjust for potential factor differences at baseline. The analyses of secondary outcomes for evaluating the initial timing (h) of the audit and monitoring intervention by the ICT pharmacists were as follows: Analysis for death within 30 days was conducted using multivariable logistic regression analysis adjusted by age, sex, eCCr, albumin, comorbidities (cardiovascular diseases), ICU admission, the loading dose of vancomycin, and interventions of clinical pharmacists in charge of the wards at baseline; analysis for implementation of de-escalation was conducted using multivariable logistic regression analysis adjusted by age, sex, comorbidities (diabetes, cancer, cardiovascular diseases, COPD, hepatic disease, renal failure, hypertension, and dyslipidaemia), immunosuppressant use, ICU admission, and interventions of clinical pharmacists in charge of the wards at baseline; analysis for number of days of vancomycin administration was conducted using multiple linear regression analysis adjusted by age, sex, comorbidities (diabetes, cancer, cardiovascular diseases, COPD, hepatic disease, renal failure, hypertension, and dyslipidaemia), immunosuppressant use, ICU admission, and interventions of clinical pharmacists in charge of the wards at baseline. Subgroup analyses were performed in the strata of paediatric patients, ICU patients, non-ICU patients, and non-haematology patients.” (p12-13, line 220-244).

“Moreover, explorative analyses of primary and secondary outcomes and the subgroups were performed using the timing of intervention initiation categorised into the four groups (<24 hours, 24 to 72 hours, 72 to 120 hours, ≥120 hours) as explanatory variables.” (p14, line 249-252).

Comment 4: Authors should review manuscript grammatically and repetition of sentences throughout the manuscript should be discouraged. It is suggested to improve the sentence structures throughout the manuscript as well.

Response: Thank you for pointing this out. We re-read the entire manuscript and corrected the grammar and duplicated expressions throughout.

Comment 5: Authors should describe the concentration related adverse effects of the Vancomycin particularly in organ compromised patients (kidney/liver disease).

Response: Thank you for your comment. We have accordingly modified the Discussion section as follows:

“In this study, 51.9% of the patients failed to maintain vancomycin trough blood concentrations in the range of 10 to 20 μg/ml, and 6.9% of all patients had renal dysfunction or were suspected to be affected by or renal dysfunction due to vancomycin. The frequency of acute kidney injury by vancomycin in previous studies in patients with baseline serum creatinine levels below 2.0 mg/dl was 5%; for concentrations <10 μg/ml, 21%; for 10 to 15 μg/ml, 20%; for 15 to 20 μg/ml; and for >20 μg/ml, 33% [34]. Although 20.2% of the patients in this study had abnormal renal function at baseline, the frequency of developing renal dysfunction was not higher than that reported in previous studies, and the intervention methods of ICT pharmacists in this study may not have worsened renal function. Although we adjusted for variables reflecting renal and hepatic function when conducting the multivariate analysis, it was possible that these variables may have fluctuated after the initiation of vancomycin treatment. Further analysis over time may be necessary to verify the association between strict vancomycin concentration trends and laboratory values affecting these concentrations.” (p32, line 467-480).

Additional reference:

34. Lodise TP, Patel N, Lomaestro BM, Rodvold KA, Drusano GL. Relationship between initial vancomycin concentration-time profile and nephrotoxicity among hospitalized patients. Clin Infect Dis. 2009;49: 507-514. https://doi.org/10.1086/600884.

Comment 6: Heading of sample size should be in discussed in methodology section instead of after statistical analyses. 

[Comment 6-1]

Response: Thank you for your comment. We have accordingly modified. (p.11, line 203-214).

Discussion heading is enough, no need to write subheadings as key results and interpretations. 

[Comment 6-2] Response: Thank you for your comment. As stated, those subheadings have been removed from the Discussion section. 

Discussion section is poorly written and must be improved by addition of logical justification of the results.

[Comment 6-3] Response: Thank you for your valuable feedback. We have revised the Discussion section to explain our findings in a logical manner and have included comparisons with previous studies. 

 In addition; the safety profile of the drug (Vancomycin) should also be discussed in discussion section

[Comment 6-4]

Response: Thank you for your comment. We described the safety profile in the Discussion section.

“A possible side effect of vancomycin was skin rash or allergy (4.7%). Although common side effects include phlebitis and ototoxicity, they were not reported in this study and did not affect treatment performance. However, ototoxicity is often difficult to detect and may require careful monitoring in older adult patients who are at high risk.” (p32, line 480-483).

Comment 7: Conclusion needs improvement and should be outcome based; sentence related to safety of interventions in conclusion section is inappropriate.

Response: We agree with your comment and have accordingly revised the Conclusion section.

“The initial timing of the comprehensive audit and monitoring intervention by ICT pharmacists did not affect the maintenance of the target blood concentration of vancomycin. However, an earlier start of the intervention was associated with a time-dependent reduction in the number of days of vancomycin administration.” (p.34, line520-523).

Comment 8: Authors should add the Future perspective of the study. Manuscripts lack the implications of findings of the study. It would be more appropriate if authors add its clinical significance.

Decision: Manuscript requires major revision.

Response: Thank you for your comments. We have added the future perspective at the end of the Discussion section as follows:

“The finding of this study showed that the number of days of administration tended to increase with any delay in the timing of early ICT pharmacist intervention. The intervention methods of ICT pharmacists may ensure the safety of vancomycin administration, as there were reports that showed a progressive increase in nephrotoxicity with an increase in the duration of vancomycin treatment [35]. Moreover, the safety of the intervention may be ensured, as no increase in mortality was observed. However, as audits and monitoring are time-consuming and labour-intensive, it is important to maintain IPCs to combat AMR in healthcare facilities while ensuring the availability of healthcare resources. The 2020 guideline has recommended area under the concentration time curve (AUC)-based dosing for the efficacy and safety of vancomycin, and the 2022 Japanese guidelines have also been revised [36, 37]. To adhere to strict blood collection times for the accurate assessment of vancomycin concentrations, and implement the treatment as early as possible considering the healthcare resources, future perspective strategies may need to be explored, so as to further strengthen collaboration for shared specialised knowledge between ICT pharmacists and general ward pharmacists.” (p33-34, line 503-517).

Additional references:

35. van Hal SJ, Paterson DL, Lodise TP. Systematic review and meta-analysis of vancomycin-induced nephrotoxicity associated with dosing schedules that maintain troughs between 15 and 20 milligrams per liter. Antimicrob Agents Chemother. 2013;57: 734-744. https://doi.org/10.1128/AAC.01568-12.

36. Rybak MJ, Le J, Lodise TP, Levine DP, Bradley JS, Liu C, et al. Therapeutic monitoring of vancomycin for serious methicillin-resistant Staphylococcus aureus infections: A revised consensus guideline and review by the American Society of Health-System Pharmacists, the Infectious Diseases Society of America, the Pediatric Infectious Diseases Society, and the Society of Infectious Diseases Pharmacists. Am J Health Syst Pharm. 2020;77: 835-864. https://doi.org/10.1093/ajhp/zxaa036.

37. Matsumoto K, Oda K, Shoji K, Hanai Y, Takahashi Y, Fujii S, et al. Clinical practice guidelines for therapeutic drug monitoring of vancomycin in the framework of model-informed precision dosing: A Consensus Review by the Japanese Society of Chemotherapy and the Japanese Society of Therapeutic Drug Monitoring. Pharmaceutics. 2022;14. https://doi.org/10.3390/pharmaceutics14030489.

Reviewer #4: Hideki Sugita, et al., carried out a retrospective observational study to investigate the initial timing of infection control pharmacist-driven audit and monitoring of vancomycin therapy in patients with infectious diseases. To the best of my knowledge, the study has substantial scientific merit and reads quite well. The experiments, and statistical analyses performed were according to standard and were described in sufficient detail and meets all applicable standards for the ethics of experimentation and research integrity. The limitations of the study were also highlighted. There however some minor grammatical/typographical errors. I will suggest the manuscript is thoroughly revised to correct the grammatical/typographical errors.

Response: Thank you so much for your comments. We revised our manuscript to correct grammatical and typographical errors, and to take into account the feedback from other reviewers.

Reviewer #5: The manuscript presents a retrospective observational study that evaluates the initial timing of infection control pharmacist-driven audit and monitoring of vancomycin therapy in patients with infectious diseases.

Flowchart representing the patient selection process should be provided.

[Comment 1-1]

Response: Thank you for your indication. We revised Figure 1 and the Results: Participants section (Fig 1 and p15, line 271-274). We also revised the explanatory text for Fig 1, as noted by other reviewers (p15, line 276-279).

Clinical implications of the study's findings and how they can contribute to improving patient outcomes should be discussed

[Comment 1-2]

Response: Thank you for your comments. We have added the future perspective at the end of the Discussion section as follows:

“The finding of this study showed that the number of days of administration tended to increase with any delay in the timing of early ICT pharmacist intervention. The intervention methods of ICT pharmacists may ensure the safety of vancomycin administration, as there were reports that showed a progressive increase in nephrotoxicity with an increase in the duration of vancomycin treatment [35]. Moreover, the safety of the intervention may be ensured, as no increase in mortality was observed. However, as audits and monitoring are time-consuming and labour-intensive, it is important to maintain IPCs to combat AMR in healthcare facilities while ensuring the availability of healthcare resources. The 2020 guideline has recommended area under the concentration time curve (AUC)-based dosing for the efficacy and safety of vancomycin, and the 2022 Japanese guidelines have also been revised [36, 37]. To adhere to strict blood collection times for the accurate assessment of vancomycin concentrations, and implement the treatment as early as possible considering the healthcare resources, future perspective strategies may need to be explored, so as to further strengthen collaboration for shared specialised knowledge between ICT pharmacists and general ward pharmacists.” (p33-34, line 503-517).

Additional references:

35. van Hal SJ, Paterson DL, Lodise TP. Systematic review and meta-analysis of vancomycin-induced nephrotoxicity associated with dosing schedules that maintain troughs between 15 and 20 milligrams per liter. Antimicrob Agents Chemother. 2013;57: 734-744. https://doi.org/10.1128/AAC.01568-12.

36. Rybak MJ, Le J, Lodise TP, Levine DP, Bradley JS, Liu C, et al. Therapeutic monitoring of vancomycin for serious methicillin-resistant Staphylococcus aureus infections: A revised consensus guideline and review by the American Society of Health-System Pharmacists, the Infectious Diseases Society of America, the Pediatric Infectious Diseases Society, and the Society of Infectious Diseases Pharmacists. Am J Health Syst Pharm. 2020;77: 835-864. https://doi.org/10.1093/ajhp/zxaa036.

37. Matsumoto K, Oda K, Shoji K, Hanai Y, Takahashi Y, Fujii S, et al. Clinical practice guidelines for therapeutic drug monitoring of vancomycin in the framework of model-informed precision dosing: A Consensus Review by the Japanese Society of Chemotherapy and the Japanese Society of Therapeutic Drug Monitoring. Pharmaceutics. 2022;14. https://doi.org/10.3390/pharmaceutics14030489.

6. PLOS authors have the option to publish the peer review history of their article (what does this mean?). If published, this will include your full peer review and any attached files.Do you want your identity to be public for this peer review? €b0For information about this choice, including consent withdrawal, please see our Privacy Policy.

Reviewer #1: No

Reviewer #2: No

Reviewer #3: Yes: €b0Mehmood Ahmad

Reviewer #4: Yes: €b0Isaac Akefe Oluwatobi

Reviewer #5: No 

Response: Thank you so much.

---

## [Decision Letter · Decision Letter 1]

22 Aug 2023

Evaluation of the initial timing of infection control pharmacist-driven audit and monitoring of vancomycin therapy in patients with infectious diseases: A retrospective observational study

PONE-D-23-11831R1

Dear Dr. Sugita,

We’re pleased to inform you that your manuscript has been judged scientifically suitable for publication and will be formally accepted for publication once it meets all outstanding technical requirements.

Kind regards,

Keiko Hosohata, Ph.D.

Academic Editor

PLOS ONE

Additional Editor Comments (optional):

Reviewers' comments:

Reviewer's Responses to Questions

**Comments to the Author**

1. If the authors have adequately addressed your comments raised in a previous round of review and you feel that this manuscript is now acceptable for publication, you may indicate that here to bypass the “Comments to the Author” section, enter your conflict of interest statement in the “Confidential to Editor” section, and submit your "Accept" recommendation.

Reviewer #1: All comments have been addressed

Reviewer #2: All comments have been addressed

Reviewer #4: All comments have been addressed

2. Is the manuscript technically sound, and do the data support the conclusions?

Reviewer #1: Yes

Reviewer #2: Yes

Reviewer #4: Yes

3. Has the statistical analysis been performed appropriately and rigorously? 

Reviewer #1: Yes

Reviewer #2: Yes

Reviewer #4: Yes

4. Have the authors made all data underlying the findings in their manuscript fully available?

Reviewer #1: Yes

Reviewer #2: (No Response)

Reviewer #4: Yes

5. Is the manuscript presented in an intelligible fashion and written in standard English?

Reviewer #1: Yes

Reviewer #2: Yes

Reviewer #4: Yes

6. Review Comments to the Author

Reviewer #1: The authors have done a good job in addressing some of the identified issues with the paper. It is now easier to follow and the outcomes are clearer. It would be nice to add a comment about how the results will impact the operations of the program. Given there was no impact of timing of audit on vancomycin levels, will that change the timing of the audit?

There are a few grammatical errors remaining - audit and feedback is written and audits and feedback in several places - this should be fixed.

Reviewer #2: (No Response)

Reviewer #4: The authors have adequately revised the manuscript and implemented the suggested changes.

I suggest the manuscript should be accepted if you feel ok to do so.

7. PLOS authors have the option to publish the peer review history of their article (what does this mean?). If published, this will include your full peer review and any attached files.

Reviewer #1: No

Reviewer #2: **Yes: **Nasir Ahmad, Pharm-D, PhD

Reviewer #4: **Yes: **Isaac Akefe Oluwatobi

---

## [Editor Report · Acceptance letter]

24 Aug 2023

PONE-D-23-11831R1 

Evaluation of the initial timing of infection control pharmacist-driven audit and monitoring of vancomycin therapy in patients with infectious diseases: A retrospective observational study 

Dear Dr. Sugita:

I'm pleased to inform you that your manuscript has been deemed suitable for publication in PLOS ONE. Congratulations! Your manuscript is now with our production department. 

Kind regards, 

on behalf of

Dr Keiko Hosohata 

Academic Editor

PLOS ONE